# Hierarchical Balance Packing: Towards Efficient Supervised Fine-tuning for Long-Context LLM

**Yongqiang Yao**[1,†], **Jinru Tan**[3,†], **Kaihuan Liang**[2,†], **Feizhao Zhang**[2], **Jiahao Hu** [2]
**Shuo Wu**[2], **Yazhe Niu** [4], **Ruihao Gong**[5,2,*], **Dahua Lin**[2], **Ningyi Xu**[1*]

[1]Shanghai Jiao Tong University    [2]SenseTime Research    [3]Central South University
[4]The Chinese University of Hong Kong    [5]Beihang University

soundbupt@gmail.com, tanjingru@csu.edu.cn, liangkaihuan@sensetime.com
zhangfeizhao, hujiahao1, wushuo1@sensetime.com, niuyazhe314@outlook.com,
gongruihao@buaa.edu.cn, dhlin@sensetime.com, xuningyi@sjtu.edu.cn

## Abstract

Training Long-Context Large Language Models (LLMs) is challenging, as hybrid training with long-context and short-context data often leads to workload imbalances. Existing works mainly use data packing to alleviate this issue, but fail to consider imbalanced attention computation and wasted communication overhead. This paper proposes Hierarchical Balance Packing (HBP), which designs a novel batch-construction method and training recipe to address those inefficiencies. In particular, the HBP constructs multi-level data packing groups, each optimized with a distinct packing length. It assigns training samples to their optimal groups and configures each group with the most effective settings, including sequential parallelism degree and gradient checkpointing configuration. To effectively utilize multi-level groups of data, we design a dynamic training pipeline specifically tailored to HBP, including curriculum learning, adaptive sequential parallelism, and stable loss. Our extensive experiments demonstrate that our method significantly reduces training time over multiple datasets and open-source models while maintaining strong performance. For the largest DeepSeek-V2 (236B) MoE model, our method speeds up the training by $2.4\times$ with competitive performance. Codes will be released at https://github.com/ModelTC/HBP.

## 1 Introduction

Large Language Models (LLMs) [1, 2, 3] have achieved state-of-the-art performance in tasks like machine translation, summarization, and code generation. However, many applications demand to process and understand long-context information [4, 5, 6], such as summarizing books, analyzing legal documents, or retaining context in multi-turn conversations. This underscores the necessity for long-context LLMs that can efficiently process long input sequences.

As mentioned in [1], incorporating long context during the Supervised Fine-Tuning (SFT) [7] stage is highly necessary. On the other hand, general short-context data is also crucial to maintain the model's general capabilities. However, this hybrid dataset composition introduces significant challenges, primarily in terms of speed and accuracy. For speed, long-context data intensifies training inefficiencies due to imbalanced workloads; For accuracy, long-context data can degrade performance on short-context tasks, affecting the model's general capabilities. These challenges hinder the efficiency and effectiveness of SFT for long-context LLMs.

The workload imbalance caused by the hybrid of long and short data arises from two main aspects: (1) within mini-batch imbalance [8], which is caused by excessive padding from randomized mini-batch

---

*Corresponding authors: gongruihao@buaa.edu.cn, xuningyi@sjtu.edu.cn;   †: equal contribution

39th Conference on Neural Information Processing Systems (NeurIPS 2025).

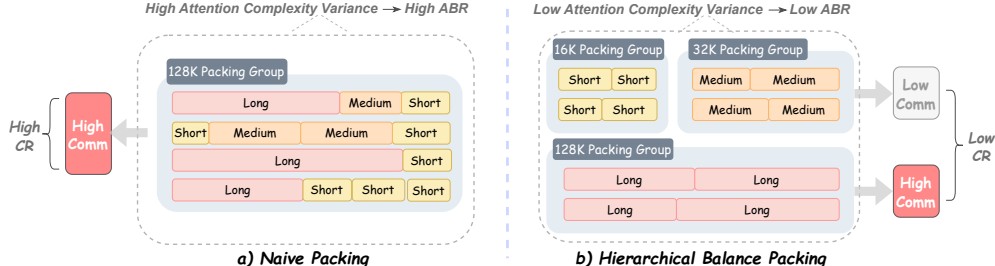

Figure 1: Difference between naive packing and hierarchical balance packing. Short, medium, and long represent different length samples, and SP Comm refers to the additional communication overhead introduced by enabling sequence parallel (SP) training. ABR (Attention Balance Ratio) measures imbalanced attention computation, and CR (Communication Ratio) measures additional communication overhead, described in Section 3.1.

construction, and (2) across mini-batch imbalance [8], which is caused by uneven computation distribution over data parallel replicas. Existing approaches mainly address this issue using data packing [6, 9], which combines variable-length data into fixed-length mini-batches. While data packing helps mitigate workload imbalances, it can also introduce new challenges. First, data packing alters the data distribution, which might affect the models' performance. Second, the complexity of attention computation for short- and long-context data differs significantly. The attention complexity of the packed samples is the sum of the attention complexities of all samples within each packing group. As illustrated in Figure 1, naive packing leads to high variance in attention complexity by directly mixing long and short samples, which causes imbalanced attention computation and workload imbalance. Third, handling long-context data requires sequential parallelism (SP) and collective communication for attention computation [10, 11], which short data do not need. When mixed, short-context data also requires sequence parallel communication, leading to wasted communication time. Larger models, such as DeepSeek-V2 (236B) [12] MoE (Mixture of Experts) models, introduce higher communication overhead due to the increased number of parameters.

To overcome the limitations of data packing, we propose Hierarchical Balance Packing (HBP), an innovative method that proposes multi-level data packing instead of conventional single-level data packing. HBP consists of three key components: (1) What are the optimal packing groups? (2) How to assign training samples to their optimal group? (3) How can a long-context LLM be trained with that data? Firstly, we propose hierarchical group auto-selection to determine the optimal packing-length group set and corresponding configurations, including the packing length, Gradient Checkpointing configuration [13], and the SP degree (how many partitions the data is divided into). Secondly, we propose balanced packing to allocate each sample to the optimal group, aiming to minimize imbalanced attention computation and communication overhead. Thirdly, we adopt alternative training between different packing groups, along with curriculum learning and a stable loss normalizer to stabilize the training process.

We validate the effectiveness of our method through extensive experiments in multiple settings. For example, on datasets Tulu3 [14] (32K) + Longcite [15] (128K), our approach speeds up **2.4×** (57.1 to 23.8 GPU Days) on DeepSeek-V2 (236B) [12]; On datasets OpenHerme [16] (4K) + Longcite (128K), our approach reduces training time from 2.95 to 2.04 GPU Days about **1.45×** speeds up on LLama3.1-8B [1]. More importantly, our method preserves performance on both short- and long-context datasets while achieving significant efficiency gains. Experiments on various models at different scales like LLama3.1-8B [1], Qwen2.5-32B [2], Qwen2.5-72B [2], and DeepSeek-V2 (236B) demonstrate consistent improvements, showing the effectiveness and generalizability of HBP.

## 2 Related Works

### 2.1 Long-Context LLM

**Long-Context Extension**. Long-Context Extensions aim to enhance LLMs' capabilities in handling long contexts. Current research can be broadly categorized into approaches that require fine-tuning

and those that operate in a zero-shot manner. Zero-shot approaches often leverage techniques such as prompt compression [17] or specially designed attention mechanisms [18, 19]. On the other hand, fine-tuning methods primarily focus on extending position encoding, such as RoPE-based approaches [5], or utilizing memory-augmented architectures [20].

**Long-Context Supervised Fine-Tuning**. For Long-Context SFT, research mainly concentrates on generating long-context datasets [21] and establishing corresponding benchmarks [22, 15, 23]. LongAlign [6] also focuses on workload balance and accuracy degradation issues. LongAlign proposed using packing and loss-reweighting to mitigate these issues. However, they failed to recognize the imbalanced attention computation and wasted communication overhead due to the packing of short- and long-context data. FlexSP [24] dynamically organizes training samples into different data groups and uses flexible sequence parallelism to enable hybrid training. Its online data organization introduces significant overhead (5-15 seconds) each iteration and fails to handle attention computation complexity across data-parallel (DP) groups. In contrast, our method introduces only negligible overhead and achieves a better computation balance of attention.

## 2.2 Data Packing

Data packing [9] is a more practical approach compared to randomly organizing data batches in LLM training. It reduces padding within batches and minimizes idle time across different data-parallel groups. Common packing methods include Random Packing [25], Sorted Batching [6], First Fit Shuffle (FFS), First Fit Decrease (FFD), Best Fit Shuffle (BFS), Shortest-Pack-First Histogram-Packing (SPFHP) [26], Iterative sampling and filtering (ISF) [8]. However, those packing methods operate on a fixed length. HBP operates data globally by introducing multiple packing groups of varying lengths, enabling more flexible and efficient handling of hybrid training with both short- and long-context data.

## 3 Problem Analysis

In this section, we first define the notations in Table 1 and introduce performance metrics in Section 3.1. We then conduct a preliminary analysis of the commonly used packing methods in Section 3.2 and training strategies in Section 3.3.

Table 1: Notation and Definitions

| Symbol | Definition |
|---|---|
| $T$ | token number in one device |
| $N$ | number of devices |
| $B$ | local batch size in one device |
| $t_i$ | token number of $i$-th sample in $B$ |
| $A$ | computation complexity of attention $\sim O(T^2)$ |
| $T_{\max}$ | maximal token number across $N$ devices |
| $A_{\max}$ | maximal attention computation across $N$ devices |
| $\text{Iter}_{\max}$ | total number of training iterations |
| $T_{\text{comm}}$ | token number for SP communication in one iteration |

Table 2: Results of ABR and CR in different sequence lengths using packing. Lower metrics indicate more efficient training.

| Packing Len | SP | ABR | CR | DBR | PR |
|---|---|---|---|---|---|
| 4K | 1 | 0.343 | 0 | 0.003 | 0.0 |
| 32K | 4 | 0.456 | 1.0 | 0.001 | 0.0 |
| 128K | 8 | 0.506 | 1.0 | 0.003 | 0.0 |

### 3.1 Measuring Metrics

**Dist Balance Ratio (DBR)** [8] quantifies the computational balance inter-devices based on length. A lower DBR 1 indicates more balanced workload distribution across different devices.

$$\text{DBR} = \frac{\sum_i^N (T_{\max} - T_i)}{T_{\max} \times N}, \quad \text{PR} = \frac{\sum_i^B (t_{\max} - t_i)}{t_{\max} \times B} \tag{1}$$

**Padding Ratio (PR)** [8] measures the proportion of wasted computations resulting from intra-device padding based on input length. A lower PR 1 indicates fewer padding tokens.

**Attention Balance Ratio (ABR)** is proposed to quantify the imbalance in attention computation for different data inter-devices. Previous metrics estimate the computational cost of attention based solely on the length of the input with full attention. However, using packing algorithms, attention computation becomes a significant factor in the overall cost. Consider the packing of a $4K$ sequence as an example, both $\{1K, 1K, 1K, 1K\}$ and $\{2K, 2K\}$ have the same total length. However, their

actual attention computation differs significantly, with complexities of $4K^2$ and $8K^2$, respectively. The Attention Balance Ratio (ABR) and example are given by 2. A lower ABR indicates a more balanced attention computation between different DP groups, resulting in less idle time.

$$\text{ABR} = \frac{\sum_i^N (A_{\max} - A_i)}{A_{\max} \times N}, \quad \text{ABR (example)} = \frac{8K^2 - 4K^2}{8K^2 \times 2} = 0.25 \tag{2}$$

**Communication Ratio (CR)** is proposed to measure the overhead of SP communication. $T_{\text{comm}_i}$ in CR 3 represents the total number of tokens that require SP communication in the current iteration, while $T_i$ denotes the total number of tokens in the current iteration. A lower CR 3 indicates that fewer tokens require SP communication, resulting in reduced communication overhead.

**Average Tokens (Ave-T)** is defined as the average number of tokens processed per iteration, serving as a measure of the model's workload. A larger Ave-T 3 indicates higher training efficiency.

$$\text{CR} = \frac{\sum_i^N T_{\text{comm}_i}}{\sum_i^N T_i}, \quad \text{Ave-T} = \frac{\sum_j^{\text{Iter}_{\max}} \left( \sum_i^N T_i \right)}{\text{Iter}_{\max} \times N} \tag{3}$$

## 3.2 Packing Analysis

**Importance of Packing**. In Table 3, we conduct three batching alternatives, random batching, ISF packing batching [8] (comparison between different packing methods is shown in Appendix A), sorted batching [6] (ensures that the sequences within each batch have similar lengths) at three sequence length 4K, 32K, 128K using Tulu3 [14] dataset. Both sorting and packing reduce DBR and PR, speeding up training. On longer sequences like 128K, packing achieves higher Ave-T than sorting, improving GPU utilization and benefiting mixed-length datasets.

**Limitation of Packing**. Although packing can partially address the efficiency issues associated with hybrid training, several limitations remain. **Imbalance Problem of Attention Complexity:** Packing samples with varying sequence lengths leads to differing attention computation complexities. Directly mixing these samples can cause imbalanced workloads and inefficient resource utilization. As illustrated in Table 2, the ABR increases significantly with the sequence length growth, indicating a rise in device idle time. More details are shown in Appendix D. **SP Communication Overhead:** long sequences require communication for attention computation, whereas short sequences do not. Directly mixing them can lead to extra communication overhead. As shown in Table 2, the CR reaches 1 when the sequence length increases, indicating that all short-context data are involved in unnecessary communication.

Table 3: Comparison of batching strategies at different sequence lengths. The local batch size $B$ is adjusted dynamically under the 32K and 128K settings in sorted batching.

| Seq Len | Batching | DBR | PR | Ave-T | GPU Days (speed up) |
|---|---|---|---|---|---|
| 4K | random | 0.540 | 0.416 | 2.4K | 8.0 (1.0×) |
| 4K | sorted | **0.001** | 0.001 | 2.4K | 3.3 (2.4×) |
| 4K | packing | 0.003 | **0.0** | **4K** | **3.1** (2.6×) |
| 32K | random | 0.64 | 0.0 | 0.8K | 16.7 (1.0×) |
| 32K | sorted | **0.01** | 0.02 | 30.2K | 4.8 (3.5×) |
| 32K | packing | 0.0007 | **0.0** | **32K** | **4.4** (3.8×) |
| 128K | random | 0.639 | 0.0 | 0.9K | 38.0 (1.0×) |
| 128K | sorted | **0.01** | 0.02 | 125K | 5.5 (6.9×) |
| 128K | packing | 0.001 | **0.0** | **128K** | **5.2** (7.3×) |

Table 4: Results of SP, minimum GC layers, and memory cost across different sequence lengths. Iter Time represents the average time taken over ten iterations.

| Seq Len | SP | GC Layer | Memory | Iter Time |
|---|---|---|---|---|
| 32K | 2 | 28 | 77G | 4.45s |
| 32K | 4 | 23 | 78G | 4.35s |
| 32K | **8** | **8** | 78G | 4.12s |
| 64K | 2 | 32 | OOM | - |
| 64K | 4 | 28 | 78G | 6.3s |
| 64K | **8** | **24** | 79G | 6.2s |
| 128K | 4 | 32 | OOM | - |
| 128K | **8** | **29** | 78G | 10.2s |
| 128K | 16 | 23 | 79G | 10.5s |

## 3.3 Training Strategy Analysis

Given the GPU resources and the data to be trained, we can select from various training strategies, provided that the VRAM requirements are satisfied. The main factors to consider are the degree of SP and the configuration of Gradient Checkpointing (GC), which is the number of layers where GC is enabled. If the SP degree is small, the VRAM demand is high, which forces an increase in

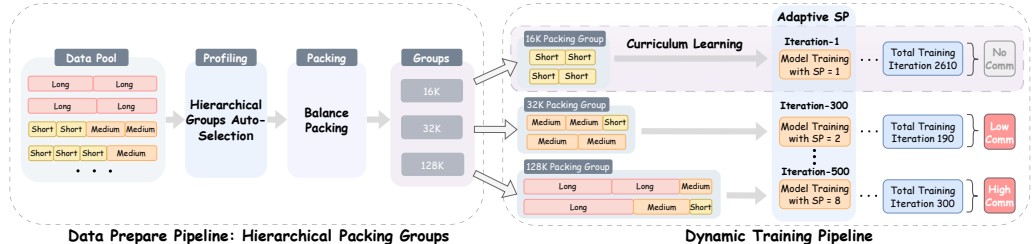

Figure 2: Hierarchical Balance Packing training framework.

the number of GC layers and leads to excessive additional computation. On the other hand, if the SP degree is large, although the VRAM demand is reduced, it introduces additional communication overhead. Therefore, there is a trade-off between the SP degree and GC configuration. Moreover, the optimal strategy varies for different sequence lengths. Table 4 shows that the optimal strategies for 32K, 64K, and 128K are different.

## 4 Hierarchical Balance Packing

In this section, we first describe how to determine optimal packing length groups (Section 4.1), then explain how each sample is assigned to a group (Section 4.2). Finally, we introduce a dynamic training pipeline for HBP (Section 4.3). The overall framework is shown in Figure 2.

### 4.1 Hierarchical Groups Auto-Selection

To determine the optimal packing length groups, we design a profile-based auto-selection algorithm as described in Algorithm 1. It operates in two stages: (1) finding the best training strategy for predefining a possible sequence length set (e.g., 8K, 16K, 32K, 64K, 128K) based on naive packing. (2) Deriving the final packing groups by optimizing communication overhead.

---

**Algorithm 1** Hierarchical Groups Auto-Selection

1: **Inputs:** Lengths $L$, Profile Time $P$, Strategy $S$
2: **Stage-1: Find the best training strategy**
3: Initialize $P \leftarrow []$, $S \leftarrow []$
4: **for** each $l \in L$ **do**
5:    $s = (sp, ckpt) \leftarrow$ FindBestSpCkpt$(l)$
6:    $P$.add(ProfileTime$(s)$),    $S$.add$(s)$
7: **end for**
8: $j \leftarrow \arg\min(P)$
9: $s_{\text{best}} \leftarrow S[j]$, $l_{\text{best}} \leftarrow L[j]$
10: $s_{\max} \leftarrow S[-1]$, $l_{\max} \leftarrow L[-1]$
11: **Stage-2: Optimize packing groups for communication**
12: $l_1 \leftarrow \lfloor l_{\text{best}}/l_{\text{best.sp}} \rfloor$,    $l_2 \leftarrow \lfloor l_{\max}/l_{\max.\text{sp}} \rfloor$
13: **if** $l_2 > l_{\text{best}}$ **then**
14:    $L_p \leftarrow [l_1, l_{\text{best}}, l_2, l_{\max}]$
15: **else**
16:    $L_p \leftarrow [l_1, l_{\text{best}}, l_{\max}]$
17: **end if**
18: **return** $L_p$

---

**Algorithm 2** Balance Packing

1: **Inputs:** Dataset $D = \{x_1, x_2, \ldots, x_n\}$, hierarchical packing groups $L_p = \{l_1, l_2, \ldots, l_n\}$
2: $G = \{G_1, G_2, \ldots, G_n\}$: packed data group
3: $B = \{B_1, B_2, \ldots, B_n\}$: final batched data
4: *GroupData*$(D, L_p)$:    splits $D$ subsets $[D_1, D_2, \ldots, D_n]$ by packing groups $L_p$.
5: *Packing $G_i =$*$(D_i, l_i)$: packing $D_i$ by $l_i$
6: *GreedyFill $G_i = (G_i, l_i, [D_{i-1}, \ldots, D_1])$*: fill data from smaller groups to reduce PR.
7: *Balance Batching*$(G_i)$: construct balance batches by sorting packed data according to attention complexity $A$ in Section 3.1.
8: ———————————————
9: Initialize $B \leftarrow []$
10: $[D_1, \ldots, D_n] \leftarrow$ GroupData$(D, L_p)$
11: **for** $i = n$ to 1 **do**
12:    $G_i \leftarrow$ Packing$(D_i, l_i)$
13:    $G_i \leftarrow$ GreedyFill$(G_i, l_i, [D_{i-1}, \ldots, D_1])$
14:    $B_i \leftarrow$ Balance Batching$(G_i)$,    $B$.add$(B_i)$
15: **end for**
16: **return** Shuffle$(B)$

---

**Stage 1: Find the best training strategy**. The algorithm begins by initializing two empty lists, $P$ and $S$, to store the profiling times and the corresponding strategies, respectively. For each possible

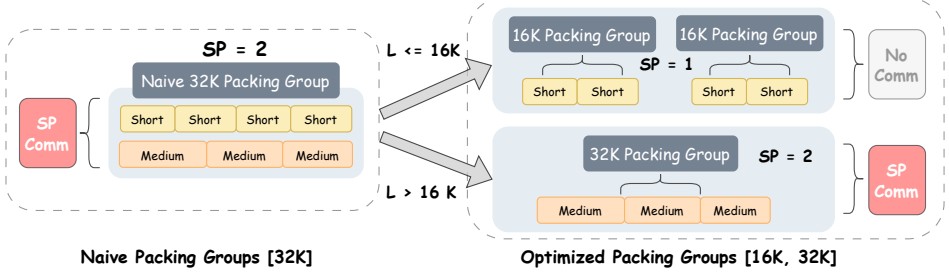

Figure 3: Example of optimizing packing group for communication.

input length $l$, we compute the optimal degree of SP and the configuration of the GC $s = (sp, ckpt)$ with the `FindBestSpCkpt` function, which achieves the optimal trade-off between the degree of SP and the configuration of the GC. More details of `FindBestSpCkpt` are shown in Appendix I.1.

Specifically, given a packing length $l$, we iterate all possible $sp$ degrees and $ckpt$ GC configurations and profile their iteration time. The best combination $(sp, ckpt)$ is then found by a greedy method. Once all input lengths are processed, the algorithm selects the best strategy by identifying the index $j$ that minimizes the profiling times in $P$. The optimal strategy and input length are:

$$j = \mathrm{argmin}(P), \quad s_{\text{best}} = S[j], \quad l_{\text{best}} = L[j] \tag{4}$$

Additionally, the maximum input length $l_{\text{max}}$ and its corresponding strategy $s_{\text{max}}$ are also retained.

**Stage 2: Optimize packing groups for communication**.

After calculating the optimal $l_{best}$ and the corresponding $s_{best}$, we can optimize it further to reduce communication overhead and obtain the final packing group lengths. Taking 32K packing groups in Figure 3 with SP=2 as an example, each SP process handles a split of 16K tokens. For samples longer than 16K, additional communication is required anyway. For samples shorter than 16K, we can group them into 16K packing groups, which is equivalent to training with 16K packing groups using SP=1, thereby reducing communication.

$$l_1 = \left\lfloor \frac{l_{\text{best}}}{s_{\text{best.sp}}} \right\rfloor \quad \text{and} \quad l_2 = \left\lfloor \frac{l_{\text{max}}}{s_{\text{max.sp}}} \right\rfloor \tag{5}$$

For $l_{\text{best}}$, the smallest packing group $l_1$ is derived based on its optimal degree of SP $s_{\text{best.sp}}$, ensuring no communication overhead during sequential parallelism. Similarly, the smallest packing group $l_2$ for $l_{\text{max}}$ is also considered. If $l_2$ is smaller than $l_{\text{best}}$, it will be merged into the $l_{\text{best}}$ range. Finally, the hierarchical packing groups $L_p = \{l_1, l_{best}, l_2, l_{max}\}$ is obtained. More implementation details are shown in Appendix I.

### 4.2 Balance Packing

After obtaining the optimal hierarchical pack groups, we distribute the dataset samples into different groups while ensuring that the metrics outlined in Section 3.1 (DBR, PR, ABR, and CR) are well-optimized, which conventional packing struggles with. We first divide the entire dataset into sub-datasets $[D_1, D_2, \ldots, D_n]$ based on $L_p$. For each $D_i$, the following steps are executed:

**(1) Packing**: We pack the data in $D_i$ to length $l_i$. This ensures low PR and DBR. Note that arbitrary packing methods are feasible.

**(2) GreedyFill**: We use the remaining unpacked data for a greedy fill, as larger groups struggle to fill the packed data using only their samples. A simple example illustration is shown in Appendix E.

**(3) Balance Batching**: Constructing balance batches $B_i$ based on attention complexity using heuristic solution sorting support maintaining balanced attention computation across different packing groups. Heuristic solutions are an efficient, approximate solution for large-scale datasets. More evidences are shown in Appendix D.

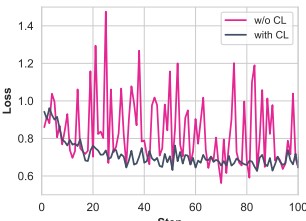
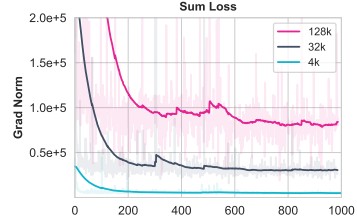
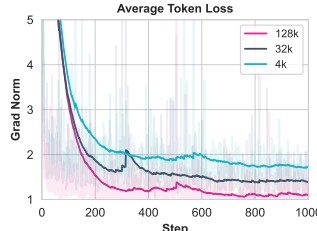

Figure 4: Loss with and without curriculum learning (CL).

Figure 5: Grad Norm of Sum loss and Average Token loss.

The procedure of balanced packing is illustrated in Algorithm 2. Since our approach inherently involves multiple levels, i.e., hierarchical packing groups, it automatically separates short- and long-context data, avoiding wasted communication overhead and imbalanced attention computation and reducing CR and ABR significantly. We also achieve low PR and DBR at multiple levels, thanks to GreedyFill. More details about the implementation are shown in Appendix J.

### 4.3 Dynamic Training Pipeline

Since HBP involves multi-level inputs, it is essential to design a dynamic training pipeline, enabling hot switching of different packing groups. Adaptive SP (pre-initialized multi-SP achieving zero overhead) is proposed to ensure efficient training with multi-level packing. Curriculum Learning and a Stable Loss Normalizer are proposed to improve the performance in hybrid training.

**Adaptive Sequential Parallel**: Each packing group is assigned an optimal training configuration $s = (\text{sp}, \text{ckpt})$, and we adopt an alternating scheme that selects the best SP degree and GC setting per packing group, as illustrated in Figure 2.

**Curriculum Learning Strategy**: Training on long-context tasks presents challenges because initiating training without instructional capabilities can result in significant fluctuations in the training loss, as illustrated in Figure 4. Thanks to our inherent hierarchical structure, it is straightforward to adopt a curriculum learning strategy that begins with general short-context data in the early stages of training. As training progresses, we transition to a hybrid approach that alternates training both short- and long-context data, as illustrated in Figure 2.

**Stable Loss Normalizer**: The training stability introduced by data packing is an important problem, as it impacts the data distribution. Previous work [6] has analyzed loss calculation, identifying two primary loss normalizers $\mathcal{L}_{\text{token}}$ (Token-Mean) and $\mathcal{L}_{\text{sample}}$ (Sample-Mean):

$$\mathcal{L}_{\text{token}} = \frac{\sum_i^{B_l} \text{loss}_i}{\sum_i^{B_l} T_i}, \quad \mathcal{L}_{\text{sample}} = \frac{\sum_i^{B_l} \frac{\text{loss}_i}{T_i}}{B_l}, \quad T_{\text{ave}} = \frac{\sum_i^{B_g} T_i}{B_g}, \quad \mathcal{L}_{\text{stable}} = \frac{\sum_i^{B_l} \text{loss}_i}{B_l * T_{\text{ave}}} \quad (6)$$

where $B_l$ represents the local batch size within the DP group, and $T_i$ denotes the number of loss tokens in the batch $i$. They argued that instability lies in the inconsistent loss of normalization. [27] also proposes sum loss without normalization to mitigate the effect of norm discrepancies. However, the sum loss introduces a trade-off: as the sequence length increases, the gradient values escalate disproportionately (**1e+5**), as shown in the left part of Figure 5. To address the above issues, we empirically observe $T_{\text{ave}}$ (Average Token) of the global batch size $B_g$ across all DP groups can serve as a **stable loss normalizer**. The stable loss normalizer is derived from theoretical considerations, with the objective of guaranteeing that each token contributes equally to the aggregate loss. This formulation mitigates biases introduced by heterogeneous sequence lengths, varying data-parallel group sizes, and gradient accumulation strategies. We illustrate the effect of different normalization strategies under a simple setting: a data-parallel (DP) group of size 2, where each rank processes a local batch of 2 samples.

**Token-Mean**: In the token-mean approach, the loss for each DP rank is normalized by its local token:

$$\mathcal{L}_{\text{dp}_1} = \frac{\text{loss}_1 + \text{loss}_2}{T_1 + T_2}, \quad \mathcal{L}_{\text{dp}_2} = \frac{\text{loss}_3 + \text{loss}_4}{T_3 + T_4}$$

The global loss is then averaged across ranks:

$$\mathcal{L}_{\text{final}} = \frac{\mathcal{L}_{\text{dp}_1} + \mathcal{L}_{\text{dp}_2}}{2} = \frac{\frac{\text{loss}_1 + \text{loss}_2}{T_1 + T_2} + \frac{\text{loss}_3 + \text{loss}_4}{T_3 + T_4}}{2}$$

This formulation normalizes losses rank-wise without accounting for global token distribution, potentially introducing bias when sequence lengths vary across ranks.

**Sample-Mean**: In the sample-mean method, losses are first normalized per sample and then averaged:

$$\mathcal{L}_{\text{dp}_1} = \frac{\frac{\text{loss}_1}{T_1} + \frac{\text{loss}_2}{T_2}}{2}, \quad \mathcal{L}_{\text{dp}_2} = \frac{\frac{\text{loss}_3}{T_3} + \frac{\text{loss}_4}{T_4}}{2}$$

The global loss becomes:

$$\mathcal{L}_{\text{final}} = \frac{\mathcal{L}_{\text{dp}_1} + \mathcal{L}_{\text{dp}_2}}{2} = \frac{\frac{\text{loss}_1}{T_1} + \frac{\text{loss}_2}{T_2} + \frac{\text{loss}_3}{T_3} + \frac{\text{loss}_4}{T_4}}{4}$$

This strategy balances individual samples but fails to reflect the total token count, giving undue weight to shorter sequences.

**Stable Loss Normalizer**: To eliminate such bias, we normalize by the global average token length:

$$\mathcal{L}_{\text{ave}} = \frac{T_1 + T_2 + T_3 + T_4}{4}, \quad \mathcal{L}_{\text{dp}_1} = \frac{\text{loss}_1 + \text{loss}_2}{2 \cdot T_{\text{ave}}}, \quad \mathcal{L}_{\text{dp}_2} = \frac{\text{loss}_3 + \text{loss}_4}{2 \cdot T_{\text{ave}}}$$

Each DP's loss is then normalized by $T_{\text{ave}}$, And the final loss is:

$$\mathcal{L}_{\text{final}} = \frac{\mathcal{L}_{\text{dp}_1} + \mathcal{L}_{\text{dp}_2}}{2} = \frac{\text{loss}_1 + \text{loss}_2 + \text{loss}_3 + \text{loss}_4}{4 \cdot \mathcal{L}_{\text{ave}}} = \frac{\text{loss}_1 + \text{loss}_2 + \text{loss}_3 + \text{loss}_4}{T_1 + T_2 + T_3 + T_4}$$

This normalization ensures that the contribution of each token is equal, regardless of which sample or rank it belongs to, thereby producing an unbiased global loss.

## 5 Experiments

### 5.1 Experimental Setup

**Implementation Details**. We use large-scale datasets: Tulu3 (32K)[14] for general tasks and LongCite (128K)[15] for long-context tasks. Both have shown strong performance across various benchmarks. They also enhance the model's ability to handle input lengths from 0.1K to 128K tokens. We conduct experiments with the following models: LLaMA 3.1 [1], Qwen-2.5 [2], and DeepSeek-V2 (236B). Most models are trained on 32x H100 80GB GPUs using the DeepSpeed [28], while DeepSeek-V2 (236B) is trained with the Megatron-LM [29] with 256x H100 80G GPUs. In our experiments, we use DeepSpeed-Ulysses's [10] sequence parallelism approach, and the ring-attention [11] method is also applicable. We conducted ablation experiments using the LLaMA3.1-8B model. For the Longsite dataset, approximately 2k samples are uniformly sampled. The loss normalizer for baselines without special instructions is Token-Mean, while HBP uses Ave-Token. GPU days are the evaluation metric to estimate the total training time. We use a learning rate of 1e-5, weight decay of 0.01, and adopt AdamW as our optimizer.

**Evaluation.** We comprehensively evaluated the LLM's performance using OpenCompass [30]. For general tasks, several benchmark datasets were assessed, including MMLU [31], MMLU PRO [32], CMMLU [33], BBH [34], Math [35], GPQA Diamond [36], GSM8K [37], HellaSwag [38], Math-Bench [39], HumanEval [40], MBPP [41], IFEval [42], and Drop [43]. For long-context tasks, the evaluation including Ruler [44], NeedleBench [45], LongBench [46], and Longcite.

Table 5: Results of different models. The naive packing baseline ISF uses the Token-Mean loss normalizer. LongAlign uses their proposed loss-reweighting. AVE represents the average performance on general tasks. Deepseek-V2(236B) is trained in a 32K setting due to resource constraints.

| Model | General Tasks | | | | | | | Long Tasks | | | GPU Days |
|---|---|---|---|---|---|---|---|---|---|---|---|
| (Type) | AVE | MMLU | BBH | IFEval | Math | GSM8k | HumanEval | Ruler (32K\|128K) | LongBench | LongCite | (speed up) |
| *LLama3.1-8B* | | | | | | | | | | | |
| LongAlign-packing | 56.6 | 44.5 | 65.5 | 67.8 | 30.7 | 80.6 | 62.2 | 84.5 \| 57.5 | 46.7 | 67.8 | 5.4 (0.97×) |
| LongAlign-sorted | 57.6 | 62.7 | 65.4 | 67.8 | 32.8 | 82.2 | 61.6 | 85.8 \| 59.9 | 46.5 | 64.0 | 33.3 (0.16×) |
| ISF | 56.0 | 54.5 | 65.3 | 70.4 | 33.7 | 81.7 | 62.8 | 85.0 \| 67.4 | 44.0 | 71.6 | 5.22 (1.0×) |
| **HBP** | **58.2** | 63.0 | 67.2 | 67.7 | 33.0 | 81.9 | 63.4 | 85.6 \| 70.8 | 43.1 | 71.5 | 3.73 (1.4×) |
| *Qwen2.5-32B* | | | | | | | | | | | |
| ISF | 73.5 | 74.8 | 83.5 | 75.6 | 56.5 | 93.7 | 86.6 | 88.2 \| 59.3 | 51.0 | 60.2 | 21.3 (1.0×) |
| **HBP** | **76.2** | 76.6 | 83.6 | 76.0 | 57.1 | 94.4 | 84.2 | 88.3 \| 59.0 | 51.9 | 61.7 | 16.0 (1.33×) |
| *LLama3.1-70B* | | | | | | | | | | | |
| ISF | 72.1 | 78.9 | 83.0 | 77.6 | 44.1 | 85.3 | 76.8 | 91.8 \| 57.1 | 50.4 | 72,7 | 44.4 (1.0×) |
| **HBP** | **74.2** | 81.5 | 83.1 | 76.2 | 48.3 | 93.3 | 77.4 | 93.4 \| 57.5 | 52.2 | 75.3 | 31.1 (1.42×) |
| *DeepSeek-V2 (236B)* | | | | | | | | | | | |
| ISF | 71.8 | 76.8 | 84.0 | 71.1 | 41.3 | 89.0 | 78.6 | 86.6 \| - | 47.1 | - | 57.1 (1.0x) |
| **HBP** | **72.0** | 76.5 | 83.1 | 72.6 | 41.4 | 89.9 | 78.1 | 87.3 \| - | 50.3 | - | 23.8 (2.4×) |

Table 6: Results of HBP Components. This experiment uses the Token-Mean Loss Normalizer. `Hierarchical` indicates the enabled hierarchical packing. `Balance` refers to balance batching.

| Model | Hierarchical | Balance | ABR | CR | AVE | GPU Days (speed up) |
|---|---|---|---|---|---|---|
| LLaMA3.1-8B | | | 0.506 | 1.0 | 56.0 | 5.22 (1.0×) |
| LLaMA3.1-8B | ✓ | | 0.288 | 0.173 | 56.4 | 4.51 (1.2×) |
| LLaMA3.1-8B | ✓ | ✓ | **0.002** | **0.173** | **56.6** | **3.73** (1.40×) |
| LLaMA3.1-70B | | | 0.506 | 1.0 | 72.2 | 44.4 (1.0×) |
| LLaMA3.1-70B | ✓ | | 0.288 | 0.173 | 72.8 | 33.3 (1.25×) |
| LLaMA3.1-70B | ✓ | ✓ | **0.002** | **0.173** | **72.2** | **31.1** (1.43×) |

## 5.2 Main Results

In Table 5, we compare our method HBP with LongAlign [6] and packing method ISF [8]. We also notice that LongAlign improves the average general tasks to some extent; it sacrifices performance in long tasks *e.g.*, Ruler-128K. In contrast, our method maintains strong performance both in general short tasks and long tasks. The improvements are consistent across a wide range of model sizes, from 8B to 236B parameters. Notably, for the largest MoE model, DeepSeek-V2 (236B), our method achieves an impressive **2.4×** training speed-up, reducing training time from 57.1 to 23.8 GPU days. Full results are shown in Appendix C.

## 5.3 Ablation Results

**Importance of Hybrid Training**. Table 9 shows that both short- and long-context data are essential for maintaining general and long-text capabilities. "Longcite (8K)" includes only sequences up to 8K. All settings use naive packing. Removing long-context data (row 1) harms long-text performance, while removing short-context data (row 2) weakens the general ability. Row 3, with partial short data, confirms these effects.

**Components of HBP**. In Table 6, we show that the Attention Balance Ratio (ABR) and Communication Ratio (CR) can be reduced significantly with hierarchical packing. In particular, ABR drops from 0.506 to 0.288, and the CR drops from 1.0 to 0.173. By balancing the batching data with a similar complexity of attention computation, we have much more balanced mini-batches with low ABR, from 0.288 to 0.002. Overall, we achieve a 1.4× speedup. Similar results can be observed in the larger LLaMA-3.1-70B model.

**Curriculum Learning**. Table 7 presents the impact of curriculum learning on HBP. Starting with short tasks and gradually mixing in long-context tasks benefits training and convergence. We applied a similar curriculum strategy to the naive ISF baseline using advanced sampling, which also showed improvements. This confirms the general effectiveness of curriculum learning for long-context SFT. Notably, HBP naturally separates short and long contexts, making curriculum learning easier to apply.

**Iterations for Curriculum Learning training**. Table 8 presents detailed ablation studies across different models. As shown, even a simple curriculum learning setup, adding 100 steps of early-stage

Table 7: Results of curriculum learning (CL).

| Model | CL | AVE | LongBench |
|---|---|---|---|
| LLama3.1-8B-HBP | | 56.6 | 41.6 |
| LLama3.1-8B-HBP | ✓ | **58.2** | **43.1** |
| LLama3.1-70B-HBP | | 72.2 | 51.5 |
| LLama3.1-70B-HBP | ✓ | **74.2** | **52.2** |
| LLama3.1-8B-ISF | | 56.0 | **44.0** |
| LLama3.1-8B-ISF | ✓ | **57.4** | 43.4 |

Table 8: Results of different CL iterations.

| Model | CL-Iterations | AVE | LongBench |
|---|---|---|---|
| LLama3.1-8B-HBP | 0 | 56.6 | 41.6 |
| LLama3.1-8B-HBP | 100 | 58.0 | 43.1 |
| LLama3.1-8B-HBP | 500 | **58.2** | **43.1** |
| LLama3.1-70B-HBP | 0 | 72.2 | 51.5 |
| LLama3.1-70B-HBP | 100 | 73.8 | **52.4** |
| LLama3.1-70B-HBP | 500 | **74.2** | 52.2 |

Table 9: The importance of hybrid training.

| Dataset | Pack Len | AVE | LongBench | Ruler-128K |
|---|---|---|---|---|
| Tulu3 | 32K | 57.5 | 43.0 | 52.5 |
| Longcite | 128K | 18.8 | 16.7 | 68.5 |
| Tulu3 + Longcite(8K) | 32K | 58.6 | 43.2 | 62.1 |
| Tulu3 + Longcite | 128K | 56.0 | **44.0** | **67.5** |

Table 10: Results of different loss normalizers.

| Loss Normalizer | AVE | LongBench | Ruler-128K | Longcite |
|---|---|---|---|---|
| Sum | 56.7 | 42.5 | 65.2 | 70.5 |
| Sample-Mean | 55.5 | 42.9 | 46.1 | 70.3 |
| Token-Mean | 56.6 | 41.6 | 67.5 | 70.6 |
| **Ave-Token** | **57.6** | **43.1** | **70.8** | **71.2** |

training with short samples, already yields noticeable improvements. To ensure training stability, we ultimately adopt a setting with 500 steps of short-sample-only training in our final experiments.

**Stable Loss Normalizer**. We compared several loss normalization methods by training models using different normalizers while keeping all other training configurations consistent. As shown in Table 10, the Ave-Token loss normalizer achieved the highest performance in both general tasks, Average (AVE) 57.6, and long context tasks LongBench 43.1, Ruler-128K 70.8, and LongSite 71.2.

**Importance of Hierarchical Groups Auto-Selection**. Table 11 compares manual (row 2) and automatic (row 3) packing groups. Automatic selection of groups (based on Appendix F) achieves greater speedup, reducing training time from 4.25 to 3.73 GPU days.

Table 11: Ablation results of optimizing packing groups for communication. Groups represent the packing group lengths used during training, while SP refers to the Sequence Parallel degree.

| Model | Groups | Selection | SP | ABR | CR | AVE | LongBench | GPU Days (speed up) |
|---|---|---|---|---|---|---|---|---|
| LLama3.1-8B-ISF | [128K] | manual | [8] | 0.506 | 1.0 | 56.0 | 44.0 | 5.22 (1.0×) |
| LLama3.1-8B-HBP | [32K, 128K] | manual | [2, 8] | 0.002 | 1.0 | 57.9 | 43.2 | 4.25 (1.22×) |
| LLama3.1-8B-HBP | [16K,128K] | **auto** | [1,8] | 0.002 | 0.173 | 58.2 | 43.1 | **3.73 (1.4×)** |

**Compared with FlexSP**. As shown in Table 12, HBP achieves lower ABR, DBR, and PR than FlexSP, while maintaining a similar CR, contributing to faster training (4.35 to 3.73 GPU days), even considering only the model computation time. Furthermore, FlexSP introduces a per-batch grouping overhead (5-15 seconds), which further slows down training (see Appendix B). HBP performs global data-level operations in advance with negligible data overhead.

**Negligible Overhead**. The overhead of HBP is negligible: Auto-Selection takes $\sim 3$ minutes and Packing 5 seconds, together contributing less than $2\%$ of training time (see Appendix G).

Table 12: Ablation results with FlexSP. All metrics are calculated based on the official code. For FlexSP, we only consider the model computation time and ignore the impact of data grouping time.

| Model | SP | CL | PR | DBR | CR | ABR | AVE | LongBench | GPU Days (speed up) |
|---|---|---|---|---|---|---|---|---|---|
| LLama3.1-8B-ISF | [8] | | 0 | 0.001 | 1.0 | 0.506 | 56.0 | 44.0 | 5.22 (1.0×) |
| LLama3.1-8B-FlexSP [24] | [1,2,4,8] | | 0.04 | 0.064 | 0.171 | 0.36 | 56.1 | 41.8 | 4.35 (1.2×) |
| LLama3.1-8B-HBP | [1,8] | | 0 | 0.001 | 0.173 | 0.002 | 56.6 | 41.6 | 3.73 (1.4×) |
| LLama3.1-8B-HBP | [1,8] | ✓ | 0 | 0.001 | 0.173 | 0.002 | 58.2 | 43.1 | **3.73 (1.4×)** |

**Dataset Generalization**. To verify the generalization of our method, we conducted experiments on different datasets (OpenHermes[16], LongWriter[23]). Our method also achieves effective results on these datasets. Results are shown in the Appendix H.

## 6 Conclusion and Limitations

In this paper, we proposed Hierarchical Balance Packing (HBP), a novel strategy to address workload imbalances in long-context LLM training through multi-level data packing and a dynamic training pipeline. Due to limitations in computational resources, we have not conducted experiments on longer contexts, such as 256K or 512K. Additionally, we have not validated HBP on other post-training tasks, such as RLHF or DPO. We leave these explorations to future work.

## Acknowledgements

This work was supported by the National Key Research and Development Program of China (Grant No. 2023YFB4405102), the Postdoctoral Fellowship Program and the China Postdoctoral Science Foundation (Grant No. BX20250487), and in part by the Natural Science Foundation of Hunan Province (Grant No. 2024JJ6525).

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

# A  Packing Strategy Results

Table 13 illustrates the complexity of different packing strategies and their corresponding final training times. It also shows that the training indicators Data Balance Ratio (DBR), Padding Ratio (PR), and Attention Balance Ratio (ABR) are strongly correlated with the training time, emphasizing the effectiveness of these indicators. Based on the computational complexity of packing strategies and their training time, and accuracy performance, we ultimately selected ISF as our naive packing baseline.

Table 13: Results of different packing strategies in the training setting of 128K with hybrid data. N: Number of samples; M: Number of samples per pack; S: Maximum pack length; C: Number of iterations.

| Packing Strategy | Complexity | DBR | ABR | PR | Ave | LongBench | GPU Days (speed up) |
|---|---|---|---|---|---|---|---|
| No | - | 0.63 | 0.72 | **0** | 56.5 | 41.8 | 38.3 (1.00×) |
| Random | O(N) | 0 | 0.53 | 0.03 | 56.2 | 42.0 | 5.57 (6.87×) |
| ISF | C*O(N+M) | **0** | **0.506** | 0.01 | 56.6 | 41.6 | **5.22** (7.33×) |
| FFS | O(NM) | 0 | 0.508 | 0.01 | 56.3 | 42.0 | 5.24 (7.30×) |
| FFD | O(NM) | 0 | 0.515 | 0.01 | 56.4 | 42.4 | 5.48 (6.98×) |
| BFS | O(NM) | 0 | 0.508 | 0.01 | 56.8 | 41.7 | 5.24 (7.30×) |
| SPFHP | O(N+S$^2$) | 0 | 0.512 | 0.01 | 56.2 | 42.3 | 5.40 (7.10×) |

# B  Results of FlexSP data grouping

The experiments were conducted with a sequence length of 128K using 32 GPUs. The results for FlexSP were obtained by directly testing with the officially released code. The FlexSP data group method incurs a non-negligible overhead of 5–15 seconds per batch, which cannot always be hidden by the training time shown in Table 14 Rows 3 and 4.

Table 14: Results of FlexSP data grouping. Data Time refers to the time taken to fetch a batch after overlapping with model training. Model Time refers to the time spent on model training. Total Time represents the overall time, including both data and model processing. All times are averaged over 100 iterations.

| Model | Dataset | Group method | Data Time | Model Time | Total Time |
|---|---|---|---|---|---|
| GPT-7b | github | FlexSP | 0.25s | 15.4s | 15.6s |
| GPT-7b | CommonCrawl | FlexSP | 0.3s | 13.0s | 13.3s |
| GPT-7b | Wikipedia | FlexSP | 4.41s | 2.4s | 6.81s |
| GPT-7b | Tulu3 + Longcite | FlexSP | 3.43s | 3.02s | 6.45s |
| GPT-7b | Tulu3 + Longcite | HBP | 0 s | 2.75s | 2.75s |

# C  Full Results

Tables 15 and 16 present our complete results for general tasks and long-context tasks, respectively. These results collectively validate the effectiveness of our HBP method. The other dense models all use the GQA architecture, while Deepseek-V2 adopts the MLA structure, which involves more communication due to a larger number of K and V heads, similar to MHA. HBP effectively reduces the communication overhead, making the speedup more significant.

# D  More Details of Attention Imbalance Problem

**Imbalance of attention complexity:** As deduced from Megatron-LM paper, gives a packed sequence containing n sub-sequences with total lengths, each transformer layer computes cost can be approximated as: $24Bsh^2 + 4B \sum s_i^2$, where $s_i$ is the length of the i-th sub-sequence, B is the batch size, and h is the hidden size. Therefore, variance in attention complexity $\sum s_i^2$ directly impacts compute cost.

**Effectiveness of Balance Batching:**

Table 15: Full results of general tasks. The naive packing baseline ISF uses the Token-Mean loss normalizer. LongAlign uses their proposed loss-reweighting. AVE represents the average performance on general tasks. Deepseek-V2(236B) is trained in a 32K training setting due to resource constraints.

| Model | MMLU | BBH | IFEval | Math | GSM8k | HumanEval | mmlu_pro | cmmlu | GPQA | Drop | MBPP | hellaswag | mathbench-a | mathbench-t | AVE | GPU Days (speed up) |
|---|---|---|---|---|---|---|---|---|---|---|---|---|---|---|---|---|
| *LLama3.1-8B* | | | | | | | | | | | | | | | | |
| LongAlign-packing | 44.5 | 65.5 | 67.8 | 30.7 | 80.6 | 62.2 | 26.5 | 48.4 | 28.8 | 71.5 | 63.8 | 80.5 | 45.6 | 75.6 | 56.6 | 7.4 (0.7×) |
| LongAlign-sorted | 62.7 | 65.4 | 67.8 | 32.8 | 82.2 | 61.6 | 38.3 | 43.6 | 29.3 | 65.1 | 63.0 | 69.6 | 50.3 | 73.9 | 57.6 | 33.3 (0.16×) |
| ISF | 54.5 | 65.3 | 70.4 | 33.7 | 81.7 | 62.8 | 34.6 | 39.8 | 24.2 | 65.4 | 61.8 | 67.9 | 46.7 | 74.7 | 56.0 | 5.22 (1.0×) |
| HBP | 63.0 | 67.2 | 67.7 | 33.0 | 81.9 | 63.4 | 38.4 | 43.4 | 27.8 | 68.7 | 65.0 | 68.7 | 50.1 | 76.4 | 58.2 | 3.73 (1.4×) |
| *Qwen2.5-32B* | | | | | | | | | | | | | | | | |
| ISF | 74.8 | 83.5 | 75.6 | 56.5 | 93.7 | 86.6 | 59.6 | 77.6 | 37.9 | 82.7 | 80.1 | 93.5 | 64.1 | 63.5 | 73.5 | 21.33 (1.0×) |
| HBP | 76.6 | 83.6 | 76.0 | 57.1 | 94.4 | 84.2 | 59.2 | 79.5 | 41.4 | 83.5 | 80.9 | 93.5 | 70.1 | 86.2 | 76.2 | 16.00 (1.33×) |
| *LLaMA3.1-70B* | | | | | | | | | | | | | | | | |
| ISF | 78.9 | 83.0 | 77.6 | 44.1 | 85.3 | 76.8 | 55.0 | 66.9 | 41.4 | 81.9 | 76.6 | 89.3 | 65.1 | 88.0 | 72.1 | 44.40 (1.0×) |
| HBP | 81.5 | 83.1 | 76.2 | 48.3 | 93.3 | 77.4 | 60.2 | 66.3 | 43.9 | 84.1 | 78.6 | 89.0 | 67.9 | 88.4 | 74.2 | 31.10 (1.42×) |
| *Qwen2.5-72B* | | | | | | | | | | | | | | | | |
| ISF | 83.9 | 86.0 | 79.3 | 57.2 | 94.6 | 85.9 | 66.2 | 84.7 | 45.4 | 84.6 | 84.8 | 93.7 | 74.1 | 95.0 | 79.6 | 47.10 (1.0×) |
| HBP | 84.2 | 85.8 | 79.7 | 56.2 | 94.7 | 85.0 | 65.9 | 84.9 | 50.5 | 84.9 | 86.4 | 93.9 | 72.8 | 95.1 | 79.5 | 33.70 (1.40×) |
| *DeepSeek-V2 (236B)* | | | | | | | | | | | | | | | | |
| ISF | 76.8 | 84.0 | 71.1 | 41.3 | 89.0 | 78.6 | 54.2 | 76.9 | 37.9 | 77.2 | 76.7 | 90.2 | 68.4 | 92.3 | 71.8 | 57.10 (1.0×) |
| HBP | 76.5 | 83.1 | 72.6 | 41.4 | 89.9 | 78.1 | 55.5 | 73.3 | 36.4 | 78.5 | 77.4 | 90.2 | 70.1 | 92.5 | 72.0 | 23.80 (2.4×) |

Table 16: Full results of Long tasks. The naive packing baseline ISF uses the Token-Mean loss normalizer. LongAlign uses their proposed loss-reweighting. AVE represents the average performance on general tasks. Deepseek-V2(236B) is trained in a 32K training setting due to resource constraints.

| Model | Ruler (32K \| 128K) | NeedleBench (32K \| 128K) | LongBench | Longcite | GPU Days (speed up) |
|---|---|---|---|---|---|
| *LLama3.1-8B* | | | | | |
| LongAlign-packing | 84.5 \| 57.5 | 87.9 \| 85.0 | 46.7 | 67.8 | 7.4 (0.7×) |
| LongAlign-sorted | 85.6 \| 60.0 | 92.4 \| 88.9 | 46.6 | 64.0 | 33.3 (0.16×) |
| ISF | 85.0 \| 67.4 | 92.1 \| 90.1 | 44.5 | 71.6 | 5.22 (1.0×) |
| HBP | 85.6 \| 70.8 | 91.8 \| 90.0 | 43.2 | 71.5 | 3.73 (1.4×) |
| *Qwen2.5-32B* | | | | | |
| ISF | 88.2 \| 59.3 | 94.5 \| 84.6 | 51.0 | 60.2 | 21.33 (1.0×) |
| HBP | 88.3 \| 59.0 | 96.0 \| 88.9 | 51.9 | 61.7 | 16.00 (1.33×) |
| *LLaMA3.1-70B* | | | | | |
| ISF | 91.8 \| 57.1 | 95.5 \| 92.6 | 50.4 | 72.7 | 44.4 (1.0×) |
| HBP | 93.4 \| 57.5 | 95.2 \| 92.4 | 52.2 | 75.3 | 31.1 (1.42×) |
| *Qwen2.5-72B* | | | | | |
| ISF | 92.6 \| 58.5 | 94.5 \| 90.8 | 50.0 | 64.3 | 47.1 (1.0×) |
| HBP | 92.8 \| 58.2 | 95.2 \| 91.4 | 51.7 | 64.7 | 33.7 (1.40×) |
| *DeepSeek-V2 (236B)* | | | | | |
| ISF | 86.6 \| - | 96.0 \| - | 47.1 | - | 57.1 (1.0×) |
| HBP | 87.3 \| - | 95.9 \| - | 50.3 | - | 23.8 (2.4×) |

Sorting is indeed a heuristic solution, and it cannot be strictly proven mathematically to guarantee perfect balance. However, based on the results across different datasets, it achieves near-perfect balance (low ABR approaching 0).

Heuristic solutions are a common and efficient approximate solution for solving problems on large-scale datasets. Our method ensures that iterations maintain attention to balance across different datasets over 99.6% of training shown in Table 17, demonstrating the strong generalization capability of our method.

**Balance Evidence:** The left side in Figure 6 shows baseline packing with high attention cost variance across DP groups, while the right side shows our sorting-based method, which significantly reduces this variance and improves efficiency.

Table 17: ABR comparison of different datasets using HBP.

| Dataset | Packing method | ABR |
|---|---|---|
| Tulu3 | HBP | **0.001** |
| OpenHermes | HBP | **0.001** |
| Tulu3 + Longcite | HBP | **0.002** |
| OpenHermes + Longcite | HBP | **0.004** |
| Tulu3 + Longwriter | HBP | **0.003** |

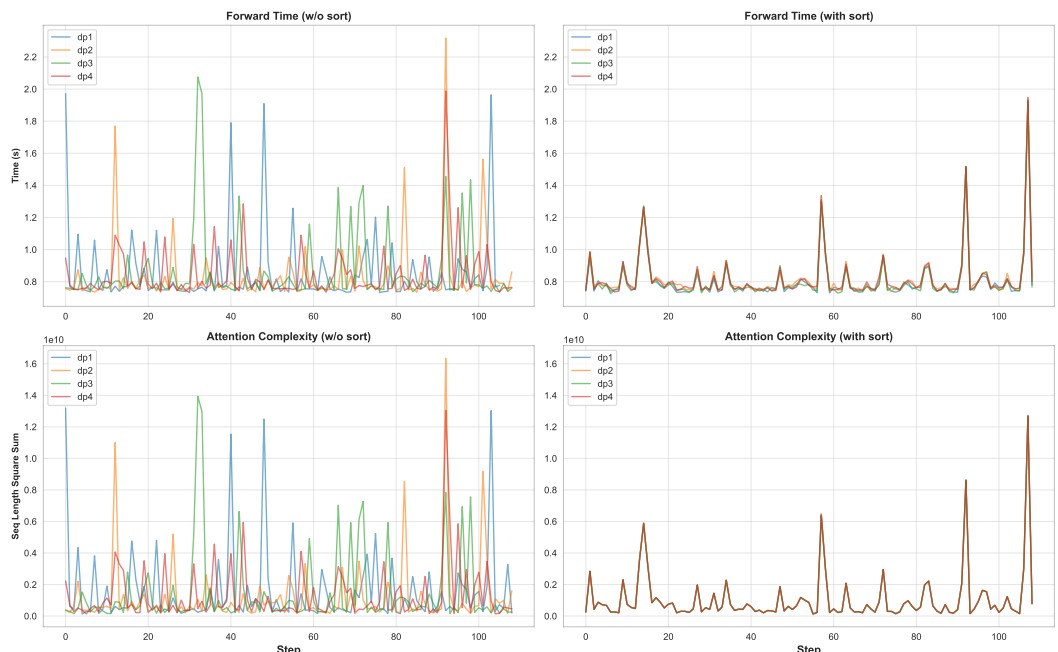

Figure 6: The plots show the forward time for different DP ranks (32 GPU DP=4, SP=8). The left column shows results without sorting, while the right column shows results after applying sort-based load balancing. The bottom plots illustrate the attention complexity (measured as the sum of sequence length squared) for each DP rank during forward passes. The top and bottom figures are used to verify the correlation between attention complexity and forward time. The left and right figures illustrate the imbalance in forward time across different DP ranks before and after applying sort-based balancing.

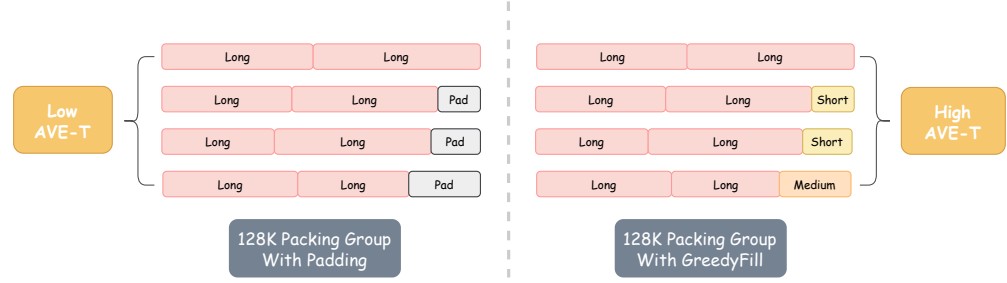

Figure 7: Example of GreedyFill. For longer samples, relying solely on data within the corresponding range (e.g., [32K, 128K]) makes it difficult to achieve full packing, resulting in low Ave-T. It operates within local packing groups by selecting a small number of sequences from other shorter packing groups to fill residual space and optimize Ave-T efficiently.

## E  More Details of GreedyFill.

GreedyFill aims to reduce intra-pack padding tokens and improve Ave-T. It operates within local packing groups by selecting a small number of sequences from other shorter packing groups to efficiently fill residual space, as shown in Figure 7.

## F  More Details of Hierarchical Groups Auto-Selection.

**The necessity of auto-selection:** Table 18 presents an example of 32K token-length training, showcasing various SP degrees and GC configurations. The second row highlights the minimal configuration $s = (2, 28)$ that satisfies memory constraints. While this configuration adheres to

memory requirements, it fails to achieve optimal performance. In contrast, the fourth row illustrates a more balanced and effective configuration with $s = (8, 8)$, achieving the fastest speed. These experiments demonstrate that, given a specific packing length, there are significant performance differences among various SP degrees and GC configurations.

**The final selection groups result:** Table 19 presents the training speed of the optimal SP degrees and GC configurations for various sequence lengths $L$. The second row of the table shows that our optimal configuration is groups = 16K, with the corresponding training strategy being $s = (1, 28)$. This indicates that the optimal training strategy is distinct for different packing groups. The evidence above emphasizes the importance of searching for the best groups and their corresponding training strategies, *i.e.,* Auto-Group Selection.

Table 18: Results of different SP degrees and GC configurations. Iter Time is the average time over ten iterations.

| SP | GC Layer | AVE | Memory | Iter Time | GPU Days |
|----|----------|------|--------|-----------|----------|
| 1 | 32 | - | OOM | - | - |
| 2 | 28 | 57.5 | 78G | 3.01 s | 3.73 |
| 4 | 23 | 57.8 | 78G | 2.95 s | 3.37 |
| 8 | 8 | 57.6 | 77G | **2.82 s** | **3.04** |
| 16 | 0 | 56.6 | 76G | 3.60 s | 3.93 |

Table 19: Results of different packing groups using hybrid data. Iter Time is the average time over ten iterations.

| Packing Groups | SP | GC Layer | Memory | Iter Time |
|----------------|-----|----------|--------|-----------|
| 8k | 2 | 8 | 77 G | 2.69 s |
| 16k | 1 | 28 | 76 G | **2.65 s** |
| 32k | 8 | 8 | 76 G | 2.83 s |
| 64k | 4 | 28 | 78 G | 3.01 s |
| 128k | 8 | 28 | 79 G | 3.05 s |

## G  Overhead Analysis

Balance Packing and Auto-Selection overhead relative to training time. Both steps together remain consistently negligible: Auto-Selection requires at most $15$ minutes and Packing only $5$ seconds, while training spans from $168$ to $1400$ minutes. Overall, the combined overhead is below $2\%$ of training time, confirming that preprocessing costs are insignificant compared to model training.

Table 20: Balance Packing and Auto-Selection overhead relative to training time.

| Model | Dataset | Auto-Selection Time | Balance Packing Time | Training Time | Overhead Ratio |
|-------|---------|---------------------|----------------------|---------------|----------------|
| LLaMA3.1-8B | Tulu3 + Longcite | **3.25 min** | **5 sec** | 168 min | **2%** |
| LLaMA3.1-70B | Tulu3 + Longcite | **15 min** | **5 sec** | 1400 min | **1%** |

### G.1  Overhead of Profiling

**Memory Profiling.** The memory profiling cost depends on the sequence length set (`Seq`) processed by the device:
$$\text{Cost}_m = \text{len}(\text{Seq}) \times \text{profile\_iter} \times \text{iteration\_time}.$$

**Time Profiling.** The total number of search strategies ($S$) is determined by the packing length set ($L$) and the Sequence Parallel ($SP$) settings. profile_iter can be adjusted based on actual iteration times, with typical values ranging from 3 to 10:
$$S = \text{len}(L) \times \text{len}(SP), \quad \text{Cost}_t = \text{len}(S) \times \text{profile\_iter} \times \text{iteration\_time}.$$

**Total Cost (Example).** For instance, with profile_iter $= 5$, $L = \{16\text{K}, 32\text{K}, 128\text{K}\}$ and $SP = \{2, 4, 8\}$, `Seq` $= \{4\text{K}, 8\text{K}, 16\text{K}\}$, for LLaMA3.1-8B:
$$\text{Cost}_t = 3 \times 3 \times 5 = 45 \times \text{iteration\_time},$$
$$\text{Cost}_m = 3 \times 5 = 15 \times \text{iteration\_time},$$
$$\text{Cost}_{\text{total}} = \text{Cost}_t + \text{Cost}_m = 60 \times \text{iteration\_time}.$$

Considering the 3000+ training iterations, totaling 60 iterations (2%) is negligible compared to the overall training time.

### G.2  Overhead of Balance Packing

For our training dataset of $1$ million samples, the overhead introduced by Balance Packing is consistently within $5$ seconds.

**Complexity.** The computational complexity of Balance Packing can be expressed as:

$$\text{Cost} = L \cdot \big( C \cdot O(N + M) + M \cdot \log M \big),$$

where $N$ is the number of samples, $M$ is the number of packing groups, $C$ is the number of ISF iterations, and $L$ denotes the selected packing set.

Since $C \ll N$, $M \ll N$, and $L \ll N$, the overall complexity reduces to a sublinear class relative to $N$. Hence, scaling to substantially larger datasets incurs only minimal additional overhead.

## G.3 Scalability of the Proposed Method

**Large-scale dataset.** Our profiling is performed for only a small number of iterations ($< 100$). As the dataset size increases, the overall overhead ratio ($< 1\%$) becomes even lower. Meanwhile, the complexity of Balance Packing is $O(N)$, so the additional cost introduced is negligible as the dataset scales.

# H  Dataset Generalization

## H.1  OpenHermes

We also provide some of our experimental results on OpenHermes. For example, Table 21 shows that the results are consistent with Tulu3 under different packing strategies. The label 22 shows that our HBP is also effective in a 128K training setting.

Table 21: Results of different packing strategies in the training setting of 4K on the OpenHermes dataset.

| Packing Strategy | Complexity | DBR | ABR | PR | Ave | LongBench | GPU Days (speed up) |
|---|---|---|---|---|---|---|---|
| - | - | 0 | 0.878 | 0.676 | 48.2 | 46.4 | 11.4 (1.0×) |
| random | O(N) | 0 | 0.648 | 0.089 | 51.44 | 47.8 | 3.64 (3.1×) |
| ISF | C*O(N+M) | 0 | 0.64 | 0.022 | 50.9 | 47.8 | 3.28 (3.5×) |
| FFS | O(NM) | 0 | 0.638 | 0.022 | 52.4 | 48.4 | 3.29 (3.5×) |
| FFD | O(NM) | 0 | 0.71 | 0.022 | 50.6 | 48.4 | 3.45 (3.3×) |
| BFS | O(NM) | 0 | 0.64 | 0.022 | 52.3 | 47.7 | 3.33 (3.4×) |
| SPFHP | O(N+S$^2$) | 0 | 0.71 | 0.029 | 52.0 | 47.8 | 3.47 (3.3×) |

Table 22: Results of HBP Components on OpenHermes + Longcite dataset. This experiment uses the Token-Mean Loss Normalizer. Balance refers to enabling Attention Balance Sort, while Hierarchical indicates the activation of hierarchical packing.

| Hierarchical | Balance | ABR | AVE | LongBench | Ruler-32K | Longsite | GPU Days (speed up) |
|---|---|---|---|---|---|---|---|
| | | 0.513 | 52.1 | 47.2 | 87.7 | 72.1 | 2.95 (1.0×) |
| ✓ | | 0.269 | 53 | 47.1 | 89.1 | 72.0 | 2.33 (1.27×) |
| ✓ | ✓ | 0.004 | 52.4 | 47.1 | 88.3 | 72.3 | 2.04 (1.44×) |

## H.2  LongWriter

Figure 8 and Table 23 present the results of our HBP hybrid training on Tulu3 and LongWriter. The x-axis represents the user instruction required length, and the y-axis represents the model output length based on the Long-Writer paper. A stronger linear correlation indicates better instruction-following capability of the model. The left side of the Figure shows the original baseline results without using the Long-Writer dataset. The middle shows the results of naive packing, and the right shows the results of HBP. The experiment demonstrates that our method is equally effective, achieving consistent acceleration across both models.

Table 23: Model Evaluation Results of Tulu3 and LongWriter Hybrid Training.

| Model | Dataset | AVe | LongBench | LongWriter | GPU Days (speed up) |
|---|---|---|---|---|---|
| Llama3.1-8B-ISF | Tulu3 + LongWriter | 57.3 | 40.9 | 67.0 | 4.4 (1.0×) |
| Llama3.1-8B-HBP | Tulu3 + LongWriter | 57.8 | 42.7 | 66.4 | 3.5 (1.26×) |

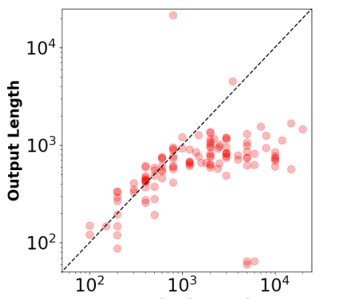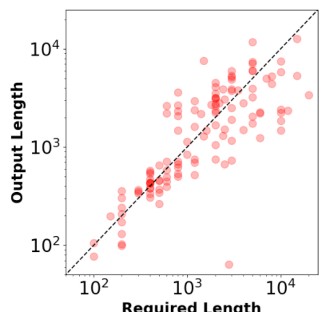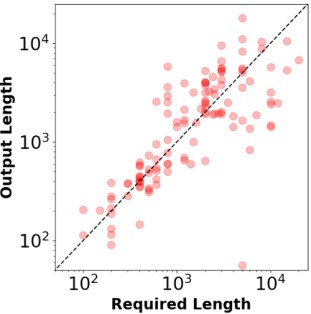

Figure 8: Results of Long-Writer Dataset for evaluating the impact of our method on accuracy. The x-axis represents the user instruction required length, and the y-axis represents the model output length based on the Long-Writer paper. A stronger linear correlation indicates better instruction-following capability of the model. The left side of the Figure shows the original baseline results without using the Long-Writer dataset. The middle shows the results of naive packing, and the right shows the results of HBP.

# I Implementation Details of Hierarchical Groups Auto-Selection

## I.1 FindBestSpCkpt

---

**Algorithm 3** FindBestSpCkpt Function

---
1: Initialize $P \leftarrow [], O \leftarrow []$
2: **for** each $sp \in SP$ **do**
3:     $ckpt \leftarrow$ GreedyProfileCkpt($l$)
4:     $P$.add(ProfileTime($l, sp, ckpt$)),
    $O$.add($sp, ckpt$)
5: **end for**
6: $j \leftarrow \text{argmin}(P)$
7: **return** $O[j]$

---

**Algorithm 4** GreedyProfileCkpt

---
1: Inputs: $l, sp, c_{min}, c_{max}$
2: $s_1 \leftarrow (sp, l, c_{min})$
3: $s_2 \leftarrow (sp, l, c_{max})$
4: $m_1^{\mathrm{r}} \leftarrow$ ProfileMemory($s_1$)
5: $m_2^{\mathrm{r}} \leftarrow$ ProfileMemory($s_2$)
6: $m_{ave} \leftarrow (m_2^{\mathrm{r}} - m_1^{\mathrm{r}})/(c_{max} - c_{min})$
7: $c_o \leftarrow c_{max} - m_2^{\mathrm{r}}/m_{ave}$
8: **return** $c_o$

---

Algorithm 3 determines the optimal gradient checkpointing strategy by evaluating all possible $sp$ strategies.

- **Initialization**: Start with empty lists $P$ and $O$ for profiling times and configurations, respectively.
- **Iterate Over Strategies**: For each strategy $sp \in SP$:
  - Compute the best gradient checkpointing configuration ($ckpt$) using the *GreedyProfileCkpt* function.
  - Use *ProfileTime* to profile the execution time for the model with the given configuration and append it to $P$.
- **Find Optimal Strategy**: Identify the index $j$ of the minimum profiling time in $P$ using $\text{argmin}(P)$.
- **Return Best Configuration**: Return the SP degree and corresponding gradient checkpointing configuration $O[j]$.

## I.2 GreedyProfileCkpt

The algorithm 4 estimates the number of gradient checkpoint layers required for a given strategy $l$ and $sp$. Here, we provide a more detailed explanation. We use the "pynvml" library to monitor GPU memory usage for memory analysis.

**Memory Profiling:**

Step 1: Given an input length L, enable Gradient Checkpointing for all layers (32 for 7B), and record the remaining GPU memory as m1.

Step 2: Given the same input length L, enable Gradient Checkpointing for only a subset of layers (e.g., 20 layers), and record the remaining GPU memory as m2.

Step 3: From the above, we can estimate the average memory saved per layer with Gradient Checkpointing: ave_m = (m1 - m2) / (32 - 20)

By running only 10 iterations, we can obtain a reliable estimate of the memory saved per layer when using Gradient Checkpointing.

**Memory And Time Consumption Report:**

For a given input length L, we set different sequence parallelism (SP) configurations. Based on the remaining GPU memory and the previously estimated ave_m, we determine the number of layers to apply Gradient Checkpointing. Then, we run 10 iterations to record the corresponding memory usage and iteration time.

**Full workflow:**

- **Initialization**: Obtain the configurations using the minimum and maximum number of gradient checkpointing layers (based on empirical observations):

$$s_1 = (sp, l, c_{min}), \quad s_2 = (sp, l, c_{max})$$

- **Memory Profiling**: Profile the remaining memory for $s_1$ ($m_1^r$) and $s_2$ ($m_2^r$).

- **Memory Slope Calculation**: Compute the average memory slope ($ave_m$) as:

$$m_{ave} = \frac{m_2^r - m_1^r}{c_{max} - c_{min}}.$$

- **Checkpointing Layer Estimation**: Estimate the required number of checkpoints ($c_o$) as:

$$c_o = c_{max} - \frac{m_2^r}{m_{ave}}.$$

## J  Implementation Details of Balance Packing

### J.1  GroupData

Given a data set $D$ and predefined hierarchical lengths $L_p$, we evaluate the length of each data set $x$ to determine the interval $(l_{i-1}, l_i)$ to which it belongs, assigning it to the corresponding group $D_i$. The detailed procedure is outlined in Algorithm 5.

### J.2  GreedyFill

For a given packing group $G_i$ with the corresponding length $l_i$, we iterate through the smaller dataset partitions $(D_{i-1}, D_{i-2}, \ldots, D_1)$ and greedily fill the group $g$ within the current $G_i$. The detailed procedure is illustrated in Algorithm 6.

### J.3  Attention Balance Sort Function

First, we calculate the attention complexity for all data within the given packing group $G$. Then, we sort the elements in $G_i$ based on their attention complexity and construct mini-batches according to the global token number requirements as shown in Algorithm 7.

**Algorithm 5** GroupData Function

1: **Inputs:** Dataset $D = \{x_1, x_2, \ldots, x_N\}$, hierarchical packing lengths $L = \{l_1, l_2, \ldots, l_n\}$
2: Initialize: $(D_1, D_2, \ldots, D_n) \leftarrow ([], [], \ldots, [])$
3: **for** $x \in D$ **do**
4:     **if** $\text{len}(x) \in (l_{i-1}, l_i]$ **then**
5:         $D_i.\text{add}(x)$
6:     **end if**
7: **end for**
8: **return** $(D_1, D_2, \ldots, D_n)$

**Algorithm 6** GreedyFill Function

1: **for** $g \in G_i$ **do**
2:     **for** $j = i - 1 \rightarrow 1$ **do**
3:         **for** $x \in D_j$ **do**
4:             **if** $\sum_{s \in g} \text{len}(s) + \text{len}(x) \le l_i$ **then**
5:                 $g.\text{add}(x)$
6:                 Remove $x$ from $D_j$
7:             **end if**
8:         **end for**
9:     **end for**
10: **end for**
11: **return** $G_i$

**Algorithm 7** Attention Balance Sort Function

1: Initialize $A \leftarrow []$
2: **for** $g \in G$ **do**
3:     $a = \sum_{x \in g} (\text{len}(x))^2$
4:     $A.\text{add}(a)$
5: **end for**
6: Sort $G$ based on $A$
7: **return** $G$

