# OpenReview forum: "Hierachical Balance Packing: Towards Efficient Supervised Fine-tuning for Long-Context LLM"
_NeurIPS.cc/2025/Conference — NeurIPS 2025 poster_

### Official Review · Reviewer_4FuH · 2025-07-02

**Clarity:** 2
**Significance:** 3
**Originality:** 3
**Rating:** 4
**Confidence:** 3

**Summary:**

This paper addresses the problem of workload imbalance when fine-tuning large language models on mixed short- and long-context data. The authors propose Hierarchical Balance Packing (HBP), which creates multiple packing-length groups, assigns each example to its optimal configuration group via "balanced packing", and employs a dynamic training pipeline (adaptive sequence-parallelism, curriculum learning, and a stable loss normalizer). Experiments on models from 8B to 236 B parameters (e.g., LLaMA3.1-8B, Qwen2.5-32B, DeepSeek-V2) and datasets (Tulu3, LongCite, OpenHermes, etc.) show that HBP significantly improves attention balance ratio and communication ratio, yielding up to 2.4$\times$ speedup with negligible or positive impact on downstream task performance.

**Questions:**

* The term "hierarchical balancing" is somewhat misleading. While the algorithm does employ multiple packing groups, it doesn’t follow a clear hierarchical (tree-like) structure. Typically, "hierarchical" implies successive merging or nesting—e.g., combining shorter-length groups into longer ones. It would help if the authors clarified why they chose this terminology and described any implicit hierarchy in their grouping process.

* Introducing curriculum learning and adaptive sequence parallelism departs from the usual random-sampling assumptions in stochastic optimization, which could, in extreme cases, impair convergence. Are there practical downsides—such as degraded performance on certain tasks, increased risk of loss spikes, or sensitivity to hyperparameters—that practitioners should be aware of?

**Ethical Concerns:**

["NO or VERY MINOR ethics concerns only"]

**Final Justification:**

The rebuttal has addressed most of my concerns, so I would like to keep my current rating.

**Limitations:**

yes

**Paper Formatting Concerns:**

No major formatting issues in this paper are noticed.

**Quality:**

3

**Strengths And Weaknesses:**

### Strengths
1. **Well motivated and Important Problem**: Data packing plays an important role in improving both the efficiency and effectiveness of LLM training. This paper presents a practical method to tackle that challenge.

2. **Novelty in Multi-Level Packing** Unlike previous single-length packing, HBP's two-stage group auto-selection (Alg. 1) and balanced packing (Alg. 2) jointly optimize packing lengths, sequence-parallel degree, and gradient-checkpoint configuration. This approach effectively takes into account Attention Balance Ratio and Communication Ratio, significantly improving both of them.

3. **Comprehensive Empirical Validation.** Experiments span small to very large models and standard general-task (MMLU, BBH, HumanEval, etc.) and long-context benchmarks (Ruler, LongBench, LongCite). HBP consistently yields 1.3–2.4× speedups while maintaining or improving average task accuracy (Table 5, 6).

### Weakness
1. **Ablation Study**: There is an inherent trade-off among Padding Ratio, Attention Balance Ratio, Communication Ratio, and Average Tokens. I strongly encourage a systematic, pairwise analysis of these metrics—ideally visualized as a Pareto front—to clarify how different packing configurations balance efficiency and performance across these objectives.

2. **Clarity**: Figures 2 and 3 would benefit from more descriptive captions that walk the reader through each step of the main algorithm. Additionally, the manuscript uses numerous abbreviations (e.g., ABR, CR, DBR, PR, SP for sequence parallelism, GC for gradient checkpointing), which should be better illustratd in the context, especially for captions of figures and tables.

---

> ### Author Rebuttal · Authors · 2025-07-26
>
> Thank you for your helpful comment.
>
> **W1**
> > Ablation Study: There is an inherent trade-off among Padding Ratio, Attention Balance Ratio, Communication Ratio, and Average Tokens. I strongly encourage a systematic, pairwise analysis of these metrics—ideally visualized as a Pareto front—to clarify how different packing configurations balance efficiency and performance across these objectives.
>
>
> **A1**
>
> We agree that a  **systematic, pairwise analysis of these metrics** of how individual metrics affect LLM training would be valuable.
>
> Since the metrics are interdependent, we did not isolate them in ablation studies. However, most related results are already shown in **Tables 3 and 6**, and we have **summarized** the experimental results from the paper into the following table to illustrate their impact more clearly. We will add this ablation study in the future version:
>
>
> |      Model      | Seq Len |        Batching        |  DBR  |  PR  |  AVT  |  CR  |  ABR  | GPU days |
> |:---------------:|:-------:|:---------------------:|:-----:|:----:|:-----:|:----:|:-----:|:--------:|
> | LLama3.1-8B     |  128K   |        random         | 0.639 | 0.0  | 0.9K  |  1   |   0.61   |   38.0   |
> | LLama3.1-8B     |  128K   |        sorted         | 0.02  | 0.0  | 125K  |  1   |   0.51   |   5.5    |
> | LLama3.1-8B     |  128K   |     lsf packing       | 0.001 |  0   | 128K  |  1   | 0.506 |   5.2    |
> | LLama3.1-8B     |  128K   |         HBP （Hierarchical）           | 0.001 |  0   | 128K  | 0.17 | 0.288 |   4.2    |
> | LLama3.1-8B     |  128K   |         HBP （Hierarchical + balance）          | 0.001 |  0   | 128K  | 0.17 | 0.002 |   3.7    |
>
> **Results Analysis**
>
> - Comparing the first and second rows, DBR drops significantly and AVT increases, reducing training time from **38 to 5.5** GPU days.
>
> - From the second to third row, AVT reaches the maximum 128K, slightly improving speed (**5.5 → 5.2 GPU days**).
>
> - The third to fourth row shows a sharp drop in CR and ABR, leading to further reduction in training time (**5.2 → 4.2 GPU days**).
>
> - Finally, the fifth row shows that reducing ABR even more translates into the lowest training cost of 3.7 GPU days.
>
> These results demonstrate how improvements in **individual metrics (e.g., DBR, AVT, ABR) correlate with more efficient LLM training**.
>
> ---
>
> **W2**
> > Clarity: Figures 2 and 3 would benefit from more descriptive captions that walk the reader through each step of the main algorithm. Additionally, the manuscript uses numerous abbreviations (e.g., ABR, CR, DBR, PR, SP for sequence parallelism, GC for gradient checkpointing), which should be better illustratd in the context, especially for captions of figures and tables.
>
> **A2**
>
> Thank you for your valuable feedback. We will revise the captions of Figures 2 and 3 to provide more detailed explanations of each step in the main algorithm. Additionally, we will improve the clarity of all abbreviations by defining them more clearly in the text and in the captions of figures and tables.
>
> ---
>
> **Q1**
> > The term "hierarchical balancing" is somewhat misleading. While the algorithm does employ multiple packing groups, it doesn’t follow a clear hierarchical (tree-like) structure. Typically, "hierarchical" implies successive merging or nesting—e.g., combining shorter-length groups into longer ones. It would help if the authors clarified why they chose this terminology and described any implicit hierarchy in their grouping process.
>
> **A3**
>
> Thank you for pointing out the potential ambiguity regarding the term “hierarchical balancing.”.
>
> Our use of the term “hierarchical” is intended to highlight how our method organizes and balances packing across multiple levels of granularity.
>
> Specifically, our approach first partitions the data at a coarse level (such as by length intervals) and then further refines the packing within each of these groups. While our method does not construct a conventional hierarchical (tree-like) structure or employ successive merging, it does involve multi-level grouping and coordinated resource allocation. In this sense, “hierarchical” reflects the layered nature of our organizational strategy rather than a strict tree-based hierarchy.
>
> We will clarify this in the revised manuscript to prevent any potential confusion.
>
> ---
>
> **Q2**
> > Introducing curriculum learning and adaptive sequence parallelism departs from the usual random-sampling assumptions in stochastic optimization, which could, in extreme cases, impair convergence. Are there practical downsides—such as degraded performance on certain tasks, increased risk of loss spikes, or sensitivity to hyperparameters—that practitioners should be aware of?
>
> **A4**
>
> Thank you for your question. In our approach, curriculum learning is used only for **a few early iterations** to stabilize training, so its impact on randomness and convergence is **minimal**.
>
> **Usual random-sampling assumptions**
>
> Moreover, HBP itself incorporates several sources of randomness:
>
> - The base packing method is **inherently stochastic**.
> - The alternating training strategy—switching between short- and long-context data—**introduces additional randomness**, which helps prevent length bias.
>
> **Robustness to Hyperparameter Settings**
>
>  As shown in Table 8, our curriculum learning strategy is **not sensitive** to hyperparameter settings and maintains consistent effectiveness.
>
> |        Model        | CL-Iterations |  AVE  | LongBench |
> |:-------------------:|:-------------:|:-----:|:---------:|
> | LLama3.1-8B-HBP     |      0        | 56.6  |   41.6    |
> | LLama3.1-8B-HBP     |     100       | 58.0  |   43.1    |
> | **LLama3.1-8B-HBP** |    500        | **58.2** | **43.1** |
>
>  In practice, we have **not observed significant degradation** in performance. Additionally, we conduct extensive experiments and evaluate the model's performance on both long and short sentence benchmarks to validate the robustness of our approach.
>
> ---
>
> We hope our response addresses your concerns, and we welcome any further questions or feedback.

---

### Official Review · Reviewer_ow9B · 2025-07-03

**Clarity:** 3
**Significance:** 2
**Originality:** 2
**Rating:** 4
**Confidence:** 3

**Summary:**

This paper proposes Hierarchical Balance Packing (HBP), a method to enhance the efficiency of supervised fine-tuning for long-context large language models. HBP mitigates workload imbalances inherent in hybrid long/short-context data training through a novel approach that combines multi-level packing groups, strategic sample allocation, and a dynamic training pipeline. Experiments show that HBP significantly accelerates training speed across various models and datasets without compromising performance.

**Questions:**

1. In Section 4.1, Algorithm 1 leverages a profile-based approach for hierarchical group selection. However, the paper does not clarify how the profiling was conducted, such as the number of iterations or evaluation settings. Could the authors provide details on the profiling procedure, and quantify the computational overhead of this process?
2. The GreedyFill step uses shorter samples to fill the packs of longer samples. Does it reintroduce the communication overhead and attention imbalance problems that HBP aims to solve?
3. Table 9 shows the importance of both short- and long-context data. Does the mixture ratio of short- and long-context data in the training set affect the performance of HBP?

**Ethical Concerns:**

["NO or VERY MINOR ethics concerns only"]

**Final Justification:**

I appreciate the authors’ response and keep my initial rating as borderline accept.

**Limitations:**

Please see the comments in Weakness and Questions.

**Paper Formatting Concerns:**

No Paper Formatting Concerns.

**Quality:**

2

**Strengths And Weaknesses:**

**Strengths**:

1. **The paper is well motivated**. The paper clearly analyzes the limitations caused by naive packing strategies in hybrid long/short context training, and proposed Hierarchical Balance Packing as an effective solution.
2. **The experiments are sufficient**. The paper validates HBP's effectiveness across multiple models of varying scales and several datasets. The ablation studies clearly demonstrate the contribution of each component of HBP to performance and efficiency, which supports the paper's core claims.
3. The paper is well-written and easy to follow.

**Weakness**
1. **The hierarchical groups auto-selection is inefficient**. It requires an additional profiling stage to determine the optimal configuration, yet the manuscript lacks a thorough discussion of the associated computational overhead (e.g., time cost) and the robustness of this process across different hardware environments.
2. The empirical findings would be strengthened by more rigorous theoretical analysis. Specifically, while the Stable Loss Normalizer proposed in Section 4.3 shows promising results, the paper does not provide a clear theoretical justification for why the Average Token strategy effectively overcomes the limitations of existing normalization methods.

---

> ### Author Rebuttal · Authors · 2025-07-26
>
> Thank you for your helpful comment.
>
> **W1 and Q1**
> > The hierarchical groups auto-selection is inefficient. It requires an additional profiling stage to determine the optimal configuration, yet the manuscript lacks a thorough discussion of the associated computational overhead (e.g., time cost) and the robustness of this process across different hardware environments.
>
> > In Section 4.1, Algorithm 1 leverages a profile-based approach for hierarchical group selection. However, the paper does not clarify how the profiling was conducted, such as the number of iterations or evaluation settings. Could the authors provide details on the profiling procedure, and quantify the computational overhead of this process?
>
> **A1**
>
> Thank you for your valuable feedback.
>
> Although our method introduces an additional profiling stage, this stage is **highly efficient**, accounting for **only 1–2%** of the total training time. Therefore, the overall profiling overhead is very low, and the framework remains **efficient**.
>
>
> **Profiling Process**
>
> The profiling process is currently provided in **Appendix I**. To make the profiling procedure clearer, we will include it in the main text in future versions.
>
> **Overhead of profiling process:**
>
> Compared to the total training time (168 min),  the overall overhead of **3.25 min** is negligible for  LLaMA3.1-8B.
> |        Setting        |       Dataset       | Profiling time | Training time | Overhead ratio |
> |:---------------------:|:-------------------:|:--------------:|:-------------:|:--------------:|
> |   LLaMA3.1-8B         | Tulu3 + longcite    |     **3.25 min**     |    168 min      |      **2 %**       |
> |   LLaMA3.1-70B        | Tulu3 + longcite    |     **15 min**       |   1400 min     |      **1 %**       |
>
> **Details of profile-based automatic optimization overhead**
>
>  * **Memory Profiling:**
>
> The memory profiling cost depends on the sequence length set (**Seq**) processed by the device.
>
>       Cost_m = len(Seq) × profile_iter × iteration_time
>
> * **Time Profiling:**
>
> The total number of search strategies (S) is determined by the packing length set (L) and the Sequence Parallel (SP) settings.
>
> Profile_iter can be adjusted based on actual iteration times, with typical values ranging from 3 to 10.
>
> Specifically:
>
>       S = len(L) x len(SP)
>       Cost_t = len(S) x profile_iter x iteration_time
>
> * **Total Cost** (Example)
>
> For instance, with a  profile_iter 5,  L =  [16K,32K,128K] and SP ([2, 4, 8]), Seq = [4K, 8K, 16K], LLama3.1-8B
>
>      Cost_t = 3 × 3 × 5 = 45 x iteration_time
>      Cost_m = 3 x 5 = 15 x iteration_time
>      Cost_total = Cost_t + Cost_m = 60 × iteration_time
>
> Considering the 3000+ training iterations,  totaling 60 iterations (**2%**) is negligible compared to the overall training time.
>
> **Robustness of this process across different hardware environments.**
>
> Our method is **hardware-agnostic** and consistently delivers good results across different hardware environments. To demonstrate this, we have included experimental results on A100 GPUs as an example.
>
> |      Model       |      Dataset      |  GPU   |AVE |   GPU days    |
> |:---------------:|:----------------:|:-----:|:--------------:|:--------------:|
> | LLama3.1-8B-ISF | Tulu3 + longcite | A100  |     56.2 | 9.4       ||
> | LLama3.1-8B-HBP | Tulu3 + longcite | A100  |   58.1  |**6.79 (1.38x)**   |
>
> ---
>
> **Q2**
> > The GreedyFill step uses shorter samples to fill the packs of longer samples. Does it reintroduce the communication overhead and attention imbalance problems that HBP aims to solve?
>
> **A2**
>
> Thank you for highlighting this point.
>
> **GreedyFill does not introduce attention imbalance problems.**
>
> As described in **Section 4.2**, the workflow is: **Packing → GreedyFill → Balance Batching**. The attention imbalance problems are primarily addressed by **Balance Batching**. Since Balance Batching is applied after GreedyFill, it does not reintroduce attention imbalance.
>
> **GreedyFill does introduce some additional communication overhead; however, it enables *faster training*.**
>
> Without the GreedyFill step, shorter samples would simply be replaced with padding,  leading to **wasted computation**. GreedyFill is specifically designed to minimize this waste, enabling **faster training overall**.
>
>
> | **Model**         | **Dataset**           | **GreedyFill** | **GPU days** |
> |:-----------------:|:---------------------:|:--------------:|:------------:|
> | LLaMA3.1-8B-HBP   | Tulu3 + longcite      | No           | 3.92         |
> | LLaMA3.1-8B-HBP   | Tulu3 + longcite      | ✔              | **3.73**         |
>
> ---
>
> **Q3**
> > Table 9 shows the importance of both short- and long-context data. Does the mixture ratio of short- and long-context data in the training set affect the performance of HBP?
>
> **A3:**
>
> This is an interesting question.  In our current experiments, we follow a typical SFT setting where short-context samples dominate. For example, LLaMA3 adopts a similar distribution. In our setup, the token-level ratio of short-to-long-context samples is approximately 3:2.  We have not conducted an extensive study on different short/long-context ratios in the paper. However, we performed a **small-scale experiment** by reducing the proportion of long-context samples.
>
> **Impact of Short/Long-Context Ratio**
>
> As shown in the results, the model’s performance remains stable (similar to Table 9), suggesting that HBP is robust to variations in the mixture ratio.
>
> | **Model**          | **Dataset**             | **Short_context / long_context ratio** | **AVE** | **LongBench** | **Ruler-128K** |
> |:------------------:|:-----------------------:|:-----------------------------:|:-------:|:-------------:|:--------------:|
> | LLaMA3.1-8B-HBP     | Tulu3                   | 3:0                           | 57.9    | 43.2          | 53.5           |
> | LLaMA3.1-8B-HBP     | Tulu3 + longcite-half     | 3:1                           | 58.0    | 43.5          | 70.5           |
> | LLaMA3.1-8B-HBP     | Tulu3 + longcite-full     | 3:2                           | 58.2    | 43.1          | 70.8           |
>
> ---
>
> **W2**
> > The empirical findings would be strengthened by more rigorous theoretical analysis. Specifically, while the Stable Loss Normalizer proposed in Section 4.3 shows promising results, the paper does not provide a clear theoretical justification for why the Average Token strategy effectively overcomes the limitations of existing normalization methods.
>
>
> **A4**
>
> Thank you for your valuable feedback. We will clarify this in detail and will add it in the future version.
>
> **Stable Loss Normalizer is not empirical**
>
> The Stable Loss Normalizer is designed based on theoretical principles. Its core purpose is to **ensure each token contributes equally to the overall loss, removing biases from sequence length, data-parallel size, and gradient accumulation**. Further analysis supporting this design is provided below.
>
> **In-depth analysis**
>
> To clarify, let’s consider a data-parallel (dp) group with a size of 2, where each rank runs with a local batch size of 2.
>
> Let $loss$ denote the sum loss for a sample, and $T$ represent the number of tokens involved in the loss calculation.
>
>    **Token-Mean**：
>
> In the token-mean approach, we compute the loss for each DP rank independently as follows:
>
> $$
> L_{dp_{1}} = \frac{loss_{1} + loss_{2}}{T_{1} + T_{2}}, \quad L_{dp_{2}} = \frac{loss_{3} + loss_{4}}{T_{3} + T_{4}}
> $$
>
>
> The final loss for the global batch is calculated by averaging the individual DP rank losses:
> $$L_{final}=\frac{L_{dp_{1}}+L_{dp_{2}}}{2} =\frac{loss_{1}+loss_{2}}{2·(T_{1}+T_{2})} + \frac{loss_{3}+loss_{4}}{2·(T_{3}+T_{4})}$$
>
> This method may introduce bias if the data-parallel ranks have different sequence length distributions, as it normalizes the losses independently for each rank **without considering the global batch’s overall token length distribution**.
>
>  **Sample-Mean:**
>
> In the sample-mean method, losses for each DP are first normalized by their respective token lengths and then averaged across the samples:
> $$L_{dp_{1}}=\frac{\frac{loss_{1}}{T_{1}}+\frac{loss_{2}}{T_{2}}}{2} , \quad                   L_{dp_{2}}=\frac{\frac{loss_{3}}{T_{3}}+\frac{loss_{4}}{T_{4}}}{2}$$
>
> The final loss is then:
> $$L_{final}=\frac{L_{dp_{1}}+L_{dp_{2}}}{2} =\frac{loss_{1}}{4·T_{1}} + \frac{loss_{2}}{4·T_{2}}+\frac{loss_{3}}{4·T_{3}} + \frac{loss_{4}}{4·T_{4}} $$
>
> This method balances the contribution of each individual sample. However, it does not fully account for the total number of tokens in the global batch, resulting in **samples with fewer tokens having a disproportionately larger influence on the overall loss**.
>
>  **Stable Loss Normalizer**
>
> The Average Token normalizer ensures that the total number of tokens in a global batch is considered. Here, we define the average token length across all samples:
>
> $$T_{ave}   = \frac{T_{1} + T_{2} + T_{3} + T_{4}}{4}$$
>
> Each DP's loss is then normalized by  $T_{ave}$
>
> $$L_{dp_{1}}=\frac{loss_{1}+loss_{2}}{2·T_{ave}}, \quad       L_{dp_{2}}=\frac{loss_{3}+loss_{4}}{2·T_{ave}}$$
>
> The final loss is:
> $$L_{final}=\frac{L_{dp_{1}}+L_{dp_{2}}}{2} =\frac{loss_{1} + loss_{2} +loss_{3} + loss_{4}}{4·T_{ave}} = \frac{loss_{1} + loss_{2} + loss_{3} + loss_{4}}{T_{1} + T_{2} + T_{3} + T_{4}}$$
>
> This approach effectively normalizes the loss by the total number of tokens across all samples in a global batch, **ensuring that each token, regardless of which sample it belongs to, contributes equally to the final loss**.
>
> ---
>
> We hope our response addresses your concerns, and we welcome any further questions or feedback.

---

### Official Review · Reviewer_UkHx · 2025-07-03

**Clarity:** 2
**Significance:** 3
**Originality:** 3
**Rating:** 3
**Confidence:** 3

**Summary:**

In summary, the authors propose an efficient hierarchical packing algorithm for long-context SFT for LLM. The traditional packing algorithm only has one packing group and faces two problems: 1) imbalanced attention complexity, 2) Inefficient Sequence-Parallelism (SP) communications. To tackle this challenge, the proposed method divides the training dataset into multiple packing groups with different sequence lengths to save the SP communications. Within each group, Balanced Packing is performed to maintain balanced attention computation among different batches. To support this multi-length input packing groups, a curriculum learning strategy is adopted. The proposed training strategy is tested on multiple models on an H100 cluster and is shown to improve the training efficiency with improved model performance.

**Questions:**

1. Table 2 seems important to motivate the proposed method but it lacks details. How are the numbers calculated/measured? Is it a general phenomenon?

2. The Curriculum Learning Strategy seems important for the proposed training strategy but it is not introduced in detail. For example, the proposed method is applied in two different clusters, which results in a different number of packing groups with different sequence lengths. How would the curriculum learning strategy vary for these two cases?

3. It would be great to know whether the proposed method is compatible with various parallelism strategies such as DP/PP/TP/FSDP.

**Ethical Concerns:**

["NO or VERY MINOR ethics concerns only"]

**Final Justification:**

I appreciate the authors' responses which have solved most of my concerns. I have raised my scores for originality and quality. However, I am still concerned about the clarity of the paper writing. So I am leaned to keep my initial rating of this paper

**Limitations:**

Yes

**Quality:**

3

**Strengths And Weaknesses:**

Strength:
1. This paper studies an important problem in large-scale training and is well-motivated
2. The proposed solution achieves faster training speed on multiple models with improved model performance

Weakness:
1. The proposed method is purely heuristic and lacks in-depth analysis
2. Some descriptions in the paper are not clear and confusing (Please refer to Questions)
3. The proposed packing method seems to be much more complicated than the traditional packing, but such overhead is not measured in the paper
4. The proposed training framework has to come with a curriculum learning strategy due to its multi-level nature. It is unclear whether this learning strategy could lead to worse accuracy for certain models/datasets
5. Seems the proposed method is only validated on an H100 cluster. However, to show the generalizability of the proposed method, it would be better to test it on multiple cluster setups where the hardware resource is different (e.g., A100 clusters, different CPUs)
6. The proposed method is limited only to SFT and does not apply to other tasks such as PPO/GRPO.

---

> ### Author Rebuttal · Authors · 2025-07-26
>
> **W1**
> > The proposed method is purely heuristic and lacks in-depth analysis
>
> **A1**
>
> Thank you for your valuable feedback.
> The packing problem is a well-known **NP-hard problem**, and finding exact solutions is difficult for large-scale training. As a result, **heuristic or approximate methods** are commonly used in practice. Experimental results demonstrate the efficiency and effectiveness of our heuristic solutions, HBP.
>
> We have conducted an **Extensive** Depth Analysis
>
> * **Packing Problem Analysis**
> In Section 3, we provide a thorough problem analysis and utilize various metrics to quantify the limitations of existing methods (see **Tables 2, 3, and 4**).
>
> * **Stable Loss Normalizer** We provide a brief theoretical introduction and a comparison with existing methods in **Section 4.3 (see Figure 5)**. A more comprehensive and **in-depth analysis** is included at the end of the response, and we will also add it in future versions of the paper.
>
> ---
>
> **W3**
> > The proposed packing method seems to be much more complicated than the traditional packing, but such overhead is not measured in the paper
>
> **A3**:
>
> Thank you for your question. We have included some overhead details in **Appendix I**. To clarify this, we now provide a **full quantitative data analysis** and will incorporate these results into the main text in future versions.
>
> **The overhead of HBP  is negligible.**
>
> Under the setting of LLama3.1-8B model and Tulu3 dataset, the overhead  includes：
> * Hierarchical Groups Auto-Selection (**3.25 min**)
> * Balance Packing (**5 sec**)
>
> |    Setting    |      Dataset      | Auto-Selection time | Balance Packing time | Training time | Overhead ratio |
> |:-------------:|:----------------:|:------------------:|:-------------------:|:-------------:|:--------------:|
> | LLama3.1-8B   | Tulu3 + longcite |      **3.25 min**        |       **5 sec**         |    168 m      |     **2 %**        |
> | LLama3.1-70B  | Tulu3 + longcite |      **15 min**         |        **5 sec**          |   1400 m      |     **1 %**        |
>
> Compared to the total training time (168 min),  the overall overhead of **3.25 min** is negligible. Note that HBP **ONLY** operates **before** the training process, so it introduces **NO** overhead during the training process.
>
> ---
>
> **W4**
> > The proposed training framework has to come with a curriculum learning strategy due to its multi-level nature. It is unclear whether this learning strategy could lead to worse accuracy for certain models/datasets
>
> **A4**
>
> We appreciate your concern and acknowledge that this is a possibility in theory.
>
> However,  in our experiments, we did not observe **accuracy degradation** when applying the curriculum learning strategy across multiple datasets and model scales (see **Tables 5, 7, 8, and Appendix H**). We will continue to verify its effectiveness on future models and datasets.
>
> ---
>
> **W5**
> > Seems the proposed method is only validated on an H100 cluster. However, to show the generalizability of the proposed method, it would be better to test it on multiple cluster setups where the hardware resource is different (e.g., A100 clusters, different CPUs)
>
> **A5**
>
> Our method is **hardware-agnostic**. To address this concern, we rented an A100 cluster and replicated the experiments. The method remained effective, delivering a **1.38× speedup**, comparable to the gains observed on H100. We will further clarify this in the revised version.
>
> |      Model       |      Dataset      |  GPU   |AVE |   GPU days    |
> |:---------------:|:----------------:|:-----:|:--------------:|:--------------:|
> | LLama3.1-8B-ISF | Tulu3 + longcite | A100  |     56.2 | 9.4       ||
> | LLama3.1-8B-HBP | Tulu3 + longcite | A100  |   58.1  |**6.79 (1.38x)**   |
>
> ---
>
> **W6**
> >The proposed method is limited only to SFT and does not apply to other tasks such as PPO/GRPO.
>
> **A6**
>
> Thank you for highlighting this point.
> * Our method is **not limited to SFT**,  it can be applied to the Actor training in  PPO or GRPO.
>
> * HBP results with GRPO
>
> |       Model        |   Dataset    | Len | Actor-Train Iter times |
> |:-----------------:|:------------:|:-------:|:----------------:|
> | Qwen2.5-7B   | deepscaler   |  32K    |      51 s        |
> | Qwen2.5-7B-HBP    | deepscaler   |  32K    |      **39 s**       |
>
> Due to time and resource constraints, we have not explored these extensions in this paper (see **Conclusion and Limitations**).
>
> ---
>
> **Q1**
> > Table 2 seems important to motivate the proposed method but it lacks details. How are the numbers calculated/measured? Is it a general phenomenon?
>
> **A7**
>
> **All metric calculations based on Section 3:**
>
> In Table 2, we use the Tulu3 dataset and apply the ISF packing algorithm to group sequences into 4K, 32K, and 128K.
>
> - **ABR:** As shown in **Equation (2)**,  ABR is based on attention computation complexity, which scales quadratically with sequence length. A concrete example is given in Section 3.1, and its impact on computational efficiency is discussed in **Appendix E**.
>
> - **CR**, defined in **Equation (3)**, measures the proportion of tokens involved in communication. When Sequence Parallelism (SP) > 1, all tokens are involved, so CR is 1 for the packed sequences in Table 2.
>
> - **DBR and PR**: Both are computed by **Equation (1)**. After ISF packing, sequence lengths are uniform (similar length), so DBR and PR are nearly zero.
>
> **General phenomenon across different packing algorithms and datasets.**
>
> In **Appendix B**, we show that ABR, DBR, and PR exhibit similar trends across different packing algorithms. We also demonstrate these patterns in the OpenHermes dataset in **Appendix H.1**.
>
> ---
>
> **Q2**
> > The Curriculum Learning Strategy seems important for the proposed training strategy but it is not introduced in detail. For example, the proposed method is applied in two different clusters, which results in a different number of packing groups with different sequence lengths. How would the curriculum learning strategy vary for these two cases?
>
> **A8**
>
> **Curriculum learning strategy is similar to the different case.**
>
> * **Cluster Results:**
>   Results with similar CL settings on both **A100 (see A5)** and H100 show that HBP works robustly across hardware platforms.
>
> |   CL-iter |GPU   |AVE |   GPU days    |
> |:-----:|:-----:|:--------------:|:--------------:|
> |  100 |A100  |     56.2 | 9.4       |
>  |  100 |A100  |   58.1  |**6.79 (1.38x)**   |
>
> * **Detail of Curriculum Learning:**
>   Described in **Section 4.3**. We start with short sequences, then alternate short and long sequences in later training stages.
>
> * **Hyperparameter Robustness:**
>   As shown in **Table 8**, our strategy is robust to different hyperparameter settings, packing groups, and sequence lengths.
>
> ---
>
> **Q3**
> > It would be great to know whether the proposed method is compatible with various parallelism strategies such as DP/PP/TP/FSDP.
>
> **A9**
>
> **Our method is fully compatible with DP, PP, TP, and FSDP**
>
> Our largest model, Deepseek-V2 (236B), uses **Megatron-LM 3D parallelism** (**Implementation Details in Appendix A**), and our method is **fully compatible with DP/PP/TP**. For other models, we use DeepSpeed ZeRO-3, which operates on the same principle as FSDP, so our approach is also **applicable to FSDP**.
>
> ---
>
> **Stable Loss In-depth analysis**
>
> To clarify, let’s consider a data-parallel (dp) group with a size of 2, where each rank runs with a local batch size of 2.
>
> Let $loss$ denote the sum loss for a sample, and $T$ represent the number of tokens involved in the loss calculation.
>
>    **Token-Mean**：
>
> In the token-mean approach, we compute the loss for each DP rank independently as follows:
>
> $$
> L_{dp_{1}} = \frac{loss_{1} + loss_{2}}{T_{1} + T_{2}}, \quad L_{dp_{2}} = \frac{loss_{3} + loss_{4}}{T_{3} + T_{4}}
> $$
>
>
> The final loss for the global batch is calculated by averaging the individual DP rank losses:
> $$L_{final}=\frac{L_{dp_{1}}+L_{dp_{2}}}{2} =\frac{loss_{1}+loss_{2}}{2·(T_{1}+T_{2})} + \frac{loss_{3}+loss_{4}}{2·(T_{3}+T_{4})}$$
>
> This method may introduce bias if the data-parallel ranks have different sequence length distributions, as it normalizes the losses independently for each rank **without considering the global batch’s overall token length distribution**.
>
> ---
>
>  **Sample-Mean:**
>
> In the sample-mean method, losses for each DP are first normalized by their respective token lengths and then averaged across the samples:
> $$L_{dp_{1}}=\frac{\frac{loss_{1}}{T_{1}}+\frac{loss_{2}}{T_{2}}}{2} , \quad                   L_{dp_{2}}=\frac{\frac{loss_{3}}{T_{3}}+\frac{loss_{4}}{T_{4}}}{2}$$
>
> The final loss is then:
> $$L_{final}=\frac{L_{dp_{1}}+L_{dp_{2}}}{2} =\frac{loss_{1}}{4·T_{1}} + \frac{loss_{2}}{4·T_{2}}+\frac{loss_{3}}{4·T_{3}} + \frac{loss_{4}}{4·T_{4}} $$
>
> This method balances the contribution of each individual sample. However, it does not fully account for the total number of tokens in the global batch, resulting in **samples with fewer tokens having a disproportionately larger influence on the overall loss**.
>
> ---
>
>  **Stable Loss Normalizer**
>
> The Average Token normalizer ensures that the total number of tokens in a global batch is considered. Here, we define the average token length across all samples:
>
> $$T_{ave}   = \frac{T_{1} + T_{2} + T_{3} + T_{4}}{4}$$
>
> Each DP's loss is then normalized by  $T_{ave}$
>
> $$L_{dp_{1}}=\frac{loss_{1}+loss_{2}}{2·T_{ave}}, \quad       L_{dp_{2}}=\frac{loss_{3}+loss_{4}}{2·T_{ave}}$$
>
> The final loss is:
> $$L_{final}=\frac{L_{dp_{1}}+L_{dp_{2}}}{2} =\frac{loss_{1} + loss_{2} +loss_{3} + loss_{4}}{4·T_{ave}} = \frac{loss_{1} + loss_{2} + loss_{3} + loss_{4}}{T_{1} + T_{2} + T_{3} + T_{4}}$$
>
> This approach effectively normalizes the loss by the total number of tokens across all samples in a global batch, **ensuring that each token, regardless of which sample it belongs to, contributes equally to the final loss**.
>
> ---
>
> We hope our response addresses your concerns, and we welcome any further questions or feedback.

---

> > ### Comment · Reviewer_UkHx · 2025-08-06
> > **Thanks for the rebutal**
> >
> > Thank you for the detailed responses and new experiment results. Most of my concerns have been resolved. I think this paper studies an important problem and the solution appears effective to me. The remaining concern is the generalization of this method to other training schemes like PPO/GRPO. Although the authors provide some evidence that the iteration time is faster in A6, it is unclear whether this is generally true and whether the curriculum learning will hurt the performance.  But in general I will consider raising my score. Thanks.

---

> ### Author Response · Authors · 2025-08-06
>
> Thank you very much for your valuable feedback.
>
> To clarify the remaining concern, a typical reinforcement learning (RL, such as PPO) training pipeline includes:
>
> - **Step 1**: Generate samples .
>
> - **Step 2**: Compute log-probabilities under the old policy.
>
> - **Step 3**: Compute log-probabilities under the reference policy.
>
> - **Step 4**: Update policy (actor) and value (critic) model.
>
> ---
>
> Since **Step 1** generates **variable-length (long + short)** samples, the subsequent stages (steps 2, 3, and 4) use these samples in a computation pattern highly similar to supervised fine-tuning (SFT).   As a result, they encounter batching and efficiency challenges similar to those addressed by HBP, making HBP directly applicable to these stages as well.
>
> We have already applied HBP to GRPO (**see A6**) with consistent improvements, and will further test it on more RL algorithms (PPO, REINFORCE++, GSPO) and tasks (math, code, agent) for both performance and speed to more comprehensively evaluate the generalization of HBP.
>
> Thank you again for your insightful comments.

---

### Official Review · Reviewer_no8c · 2025-07-05

**Clarity:** 3
**Significance:** 2
**Originality:** 3
**Rating:** 4
**Confidence:** 4

**Summary:**

This paper proposed a new data packaging method with automatic training configuration optimization (wrt gradient checkpointing and sequence parallelism).
The author designed several useful metrics to effectively measure the training efficiency in terms of waste of padding, waste of communication, balance of attention FLPOs and communications, etc.. These metrics could serve as helpful optimization objectives for future works as well.
The author designed automatic optimization method to decide the optimal packing length, assign data to these groups, and design the optimal training configurations for these groups.
Experiments demonstrate that the proposed method could speed up training time to 1.33-2.4x across different model sizes, while maintaining high performance on both short- and long-context tasks.

**Questions:**

1. Could the author provide more details about the overhead brought by the optimization process and discuss how scalable it is if someone considers applying it to pretraining-scale datasets?
2. Do you think clustering the data by lengths would impact the training data distribution, thus leading to very different bahaviors when different domains have very different length biases? How do you mitigate this impact?

**Ethical Concerns:**

["NO or VERY MINOR ethics concerns only"]

**Final Justification:**

The authors supplemented the additional prefilling overhead analysis as I requested and the results seem good. My concerns about the efficiency and randomness have been fully addressed with potential mitigation approaches. I have raised my score based this discussion.

**Limitations:**

yes

**Quality:**

3

**Strengths And Weaknesses:**

Strengths:
1. The set of metrics and optimization objectives is general and useful for future works to design better algorithms.
2. The proposed method improves the training efficiency without harming the performance.
3. The author identified an important but have been overlooked issue with existing data packaging approaches, and proposed a method that not only keeps the natural distribution of varied-length data but also minimizes the waste of compute and load imbalance in current training strategies.
4. Extensive experiments show that the proposed method is effective and performant.

Weaknesses:
1. The writing could be further improved--the paper used a large space to repeatedly highlight the key contributions and the limitation of the current packaging methods. Limited space was left for explaining the method and showing the results. In particular, a brief hint on what the notations stand for would be helpful than just assuming everyone to repeatedly look back at the notation table. In addition, some details might be missing, e.g., there is no explanation on what is FindBestSpCkpt function in 4.1 while both pseudocode and the main text uses it.
2. The proposed packaging method focuses only on the length distribution. However, it ignores the content distribution shift after clustering the data purely based on lengths. For example, if one domain only has short sentences, it will only appear at the early training stage, leading to catastrophic forgetting for this domain.
3. The paper uses profile-based automatic optimization, however, it did not discuss the overhead brought by this optimization process. In addition, it applies the method to Tulu3, which is a mid-scale post-training dataset. However, sequence packaging is often used in pretraining stage, where the paper lacks discussion on the scalability of the proposed method to large-scale pretraining datasets.
4. The search space only contains sequence parallelism and gradient checkpointing, while more dimensions, such as tensor parallelism, fsdp, etc., are not considered. In addition, sequence parallelism also has different categories, such as DeepSpeed version (using all-to-all communication and split over attention head) and DistFlashAttention version (using p2p communication and splitting over sequence length), while the method's optimization considers a very limited optimization space.

---

> ### Author Rebuttal · Authors · 2025-07-26
>
> Thank you for your helpful comment.
>
> **W1**
> > The writing could be further improved--the paper used a large space to repeatedly highlight the key contributions and the limitation of the current packaging methods. Limited space was left for explaining the method and showing the results. In particular, a brief hint on what the notations stand for would be helpful than just assuming everyone to repeatedly look back at the notation table. In addition, some details might be missing, e.g., there is no explanation on what is FindBestSpCkpt function in 4.1 while both pseudocode and the main text uses it.
>
> **A1:**
>
> Thank you for your valuable feedback. In future versions, we will simplify the discussion of our contributions and the limitations of existing packing methods to allocate more space to the methodology details and experimental results (**part of which are currently in the Appendix**).
>
> We will also ensure that important notations are reintroduced where necessary to improve clarity in the future version.
>
> Additionally, the "FindBestSpCkpt" function, currently described in **Appendix I**, will be relocated to the main text for improved accessibility.
>
> ---
>
> **W2 & Q2**
> > The proposed packaging method focuses only on the length distribution. However, it ignores the content distribution shift after clustering the data purely based on lengths. For example, if one domain only has short sentences, it will only appear at the
> early training stage, leading to catastrophic forgetting for this domain.
>
> >  Do you think clustering the data by lengths would impact the training data distribution, thus leading to very different bahaviors when different domains have very different length biases? How do you mitigate this impact?
>
> **A2:**
>
> Thank you for your question.
>
> **Short sentences not only appear at the early training stage**
>
> As described in **Section 4.3** and illustrated in **Figure 2**, short samples are distributed throughout the entire training stage.
> We employ an **alternating training strategy** between different packing groups with varying packing lengths (after a few iterations of Curriculum Learning)
>
> ---
>
> **How to mitigate this impact**
>
> While clustering by length can change the data distribution, the randomness and stable loss normalizer in our method greatly mitigates this effect. Our experiments (**Tables 5, 7, 10, and Appendix H Dataset Generalization**) also confirm the robustness of our approach.
>
>  * **randomness**
>     - The base packing method is inherently **stochastic**.
>     - The **alternating training strategy**—switching between short- and long-context data—introduces additional **randomness**, which helps prevent length bias.
>   * **Stable loss normalizer:** We incorporate a stable loss normalizer that ensures balanced contributions from different sentence lengths to the overall training objective.
>
> ---
>
> **W3 and Q1**
> > The paper uses profile-based automatic optimization, however, it did not discuss the overhead brought by this optimization process. In addition, it applies the method to Tulu3, which is a mid-scale post-training dataset. However, sequence packaging is often used in pretraining stage, where the paper lacks discussion on the scalability of the proposed method to large-scale pretraining datasets.
>
> >  Could the author provide more details about the overhead brought by the optimization process and discuss how scalable it is if someone considers applying it to pretraining-scale datasets?
>
> **A3:**
>
> Thank you for your question. We have included some overhead details in **Appendix I**. To clarify this, we now provide a **full quantitative data analysis** and will incorporate these results into the main text in future versions.
>
> **Overhead of optimization process:**
>
> Compared to the total training time (168 min),  the overall overhead of **3.25 min** is negligible for  LLaMA3.1-8B.
> |        Setting        |       Dataset       | Profiling time | Training time | Overhead ratio |
> |:---------------------:|:-------------------:|:--------------:|:-------------:|:--------------:|
> |   LLaMA3.1-8B         | Tulu3 + longcite    |     **3.25 min**     |    168 min      |      **2 %**       |
> |   LLaMA3.1-70B        | Tulu3 + longcite    |     **15 min**       |   1400 min      |      **1 %**       |
>
> **Details of profile-based automatic optimization overhead**
>
>  * **Memory Profiling:**
>
> The memory profiling cost depends on the sequence length set (**Seq**) processed by the device.
>
>       Cost_m = len(Seq) × profile_iter × iteration_time
>
> * **Time Profiling:**
>
> The total number of search strategies (S) is determined by the packing length set (L) and the Sequence Parallel (SP) settings. Profile_iter can be adjusted based on actual iteration times, with typical values ranging from 3 to 10.
> Specifically:
>
>       S = len(L) x len(SP)
>       Cost_t = len(S) x profile_iter x iteration_time
>
> * **Total Cost** (Example)
>
> For instance, with a  profile_iter 5,  L =  [16K,32K,128K] and SP ([2, 4, 8]), Seq = [4K, 8K, 16K], LLama3.1-8B
>
>      Cost_t = 3 × 3 × 5 = 45 x iteration_time
>      Cost_m = 3 x 5 = 15 x iteration_time
>      Cost_total = Cost_t + Cost_m = 60 × iteration_time
>
> Considering the 3000+ training iterations,  totaling **60 iterations (2%**) is negligible compared to the overall training time.
>
> **Scalability of the proposed method**
>
> *  **Large-scale dataset**
>
>      Our profiling is performed only for a small number of iterations (**<100**), so as the dataset size increases, the overall overhead ratio (**<1%**) becomes even lower. Meanwhile, the complexity of Balance Packing is an **O(N)** level, so the additional **cost introduced is negligible** as the dataset scales.
>
> * **Pretraining setting**
>
>      The pretrain setting can be used **directly**.  However, in the pre-training setup, training is typically conducted in stages:
>     The model is first trained on shorter sequences (e.g., 8K), and only in the final stage is trained on longer sequences (e.g., 128K). Both Qwen2.5 and Llama3.1 follow this phased approach, **so the issue of mixed sequence lengths during pre-training is not significant.**
>
> ---
>
> **W4**
> > The search space only contains sequence parallelism and gradient checkpointing, while more dimensions, such as tensor parallelism, FSDP, etc., are not considered. In addition, sequence parallelism also has different categories, such as DeepSpeed version (using all-to-all communication and split over attention head) and DistFlashAttention version (using p2p communication and splitting over sequence length), while the method's optimization considers a very limited optimization space.
>
> **A4:**
>
> Thank you for your valuable suggestions. We agree that the search space could be further extended.
> However, including too many dimensions would result in **high profiling overhead**. In this work, we therefore focus on the two most influential factors for long-context training—sequence parallelism and gradient checkpointing, and leave further extensions to future work.
>
> **different categories of sequence parallelism**
>
> Our method is agnostic to the specific implementation of sequence parallelism and works effectively with both commonly used SP approaches, including DeepSpeed-Ulysses and Ring-Attention. We have provided detailed implementation descriptions in Appendix A (*In our experiments, we use DeepSpeed-Ulysses’s  sequence parallelism approach, and the ring-attention method is also applicable*)
>
> |      Model            |      Dataset          |   Sequence Parallelism            |   GPU days         |
> |:--------------------:|:--------------------:|:---------------------:|:------------------:|
> | LLama3.1-8B-ISF      | Tulu3 + longcite     |     DeepSpeed-Ulysses                |   5.22 (1.0)       |
> | LLama3.1-8B-HBP      | Tulu3 + longcite     |    DeepSpeed-Ulysses  |   **3.73 (1.4x)**      |
> | LLama3.1-8B-ISF      | Tulu3 + longcite     |    Ring-Attention                |   5.3 (1.0)       |
> | LLama3.1-8B-HBP      | Tulu3 + longcite     |    Ring-Attention             |   **3.8 (1.39x)**     |
>
> ---
>
> We hope our response addresses your concerns, and we welcome any further questions or feedback.

---

> > ### Comment · Reviewer_no8c · 2025-08-05
> >
> > Thanks the authors for answering my questions. I think my questions have been fully addressed. I will raise my score.

---

> > > ### Author Response · Authors · 2025-08-06
> > > **Official Comment by Authors**
> > >
> > > Dear Reviewer,
> > >
> > > Thank you for your positive feedback, which indicates that our rebuttal has addressed your concerns. We will further clarify these points and incorporate your suggestions in the revised paper. Please let us know if you have any additional questions or comments.
> > >
> > > Best regards,
> > > The Authors

---

### Official Review · Reviewer_Nqmr · 2025-07-06

**Clarity:** 2
**Significance:** 3
**Originality:** 3
**Rating:** 4
**Confidence:** 3

**Summary:**

This paper focuses on training Long-Context Large Language Models (LLMs), with particular emphasis on data packing when mixing long-context and short-context data. The paper points out that existing data packing methods suffer from inefficiencies and imbalances. To address these issues, the paper introduces Hierarchical Balance Packing (HBP), which tackles the challenges of optimal packing group selection, data assignment, and training with packed data through three components: Hierarchical Groups Auto-Selection, Balance Packing, and Dynamic Training Pipeline. Experimental results demonstrate that HBP effectively improves training efficiency while maintaining strong performance.

**Questions:**

In Section 4.3, the paper proposes the Stable Loss Normalizer and empirically selects the Average Token method, but lacks an in-depth analysis of the underlying reasons. Furthermore, it remains unclear whether the stability advantages of the Average Token normalizer apply exclusively to HBP, or if they also generalize to other methods. Could the authors clarify this?

**Ethical Concerns:**

["NO or VERY MINOR ethics concerns only"]

**Limitations:**

yes

**Quality:**

3

**Strengths And Weaknesses:**

Strengths:
1. The paper introduces a variety of measuring metrics for data packing , providing valuable insights for future analyses of data packing strategies.
2. The paper constructs a comprehensive pipeline to address different challenges in data packing. Furthermore, the automatic packing capability of HBP reduces the difficulty of parameter tuning.
3. HBP significantly improves the training efficiency of Long-Context LLMs while maintaining strong performance across different benchmark datasets, thoroughly validating its effectiveness.
Weaknesses:
1. Section 3.1 introduces multiple measuring metrics and explains their relevance to LLM’s training, but the paper lacks experiments demonstrating how variations in individual metrics concretely affect LLM’s training performance and efficiency.
2. The paper claims that the HBP algorithm introduces only negligible overhead, but lacks detailed quantitative data showing how much time is cost by the Hierarchical Groups Auto-Selection and Balance Packing components, respectively.

---

> ### Author Rebuttal · Authors · 2025-07-26
>
> Thank you for your helpful comment.
>
> **W1**
>  > Section 3.1 introduces multiple measuring metrics and explains their relevance to LLM’s training, but the paper lacks experiments demonstrating how variations in individual metrics concretely affect LLM’s training performance and efficiency.
>
> **A1**
> We agree that a deeper analysis of how individual metrics affect LLM training would be valuable.
>
> Since the metrics are interdependent, we did not isolate them in ablation studies. However, most related results are already shown in **Tables 3 and 6**, and we have **summarized** the experimental results from the paper into the following table to illustrate their impact more clearly.
>
>
> |      Model      | Seq Len |        Batching Method  |  DBR  |  PR  |  AVT  |  CR  |  ABR  | GPU days |
> |:---------------:|:-------:|:---------------------:|:-----:|:----:|:-----:|:----:|:-----:|:--------:|
> | LLama3.1-8B     |  128K   |        random         | 0.639 | 0.0  | 0.9K  |  1   |   0.61   |   38.0   |
> | LLama3.1-8B     |  128K   |        sorted         | 0.02  | 0.0  | 125K  |  1   |   0.51   |   5.5    |
> | LLama3.1-8B     |  128K   |     lsf packing       | 0.001 |  0   | 128K  |  1   | 0.506 |   5.2    |
> | LLama3.1-8B     |  128K   |         HBP （Hierarchical）           | 0.001 |  0   | 128K  | 0.17 | 0.288 |   4.2    |
> | LLama3.1-8B     |  128K   |         HBP （Hierarchical + balance）          | 0.001 |  0   | 128K  | 0.17 | 0.002 |   3.7    |
>
> **Results Analysis**
>
> - Comparing the first and second rows, **DBR drops** significantly and **AVT increases**, reducing training time from **38 to 5.5 GPU days**.
>
> - From the second to third row, AVT reaches the maximum 128K, slightly improving speed (**5.5 → 5.2 GPU days**).
>
> - The third to fourth row shows a sharp **drop in CR and ABR**, leading to further reduction in training time (**5.2 → 4.2 GPU days**).
>
> - Finally, the fifth row shows that **reducing ABR** even more translates into the lowest training cost of **3.7 GPU days**.
>
> These results demonstrate how improvements in **individual metrics (e.g., DBR, AVT, ABR) correlate with more efficient LLM training**.
>
> **W2**
> >The paper claims that the HBP algorithm introduces only negligible overhead, but lacks detailed quantitative data showing how much time is cost by the Hierarchical Groups Auto-Selection and Balance Packing components, respectively.
>
> **A2:**
>
> Thank you for your question. We have included some overhead details in **Appendix I**. To clarify this, we now provide a full **quantitative data analysis** and will incorporate these results into the main text in future versions.
>
> **The overhead introduced by the HBP algorithm is negligible.**
>
> Under the setting of LLama3.1-8B model and Tulu3 dataset, the overhead brought by HBP includes：
> * Hierarchical Groups Auto-Selection (**3.25min**)
> * Balance Packing (**5 sec**)
>
> |    Model    |      Dataset      | Auto-Selection time | Balance Packing time | Training time | Overhead ratio |
> |:-------------:|:----------------:|:------------------:|:-------------------:|:-------------:|:--------------:|
> | LLama3.1-8B   | Tulu3 + longcite |      **3.25 min**        |        **5 sec**          |    168 min      |     **2 %**        |
> | LLama3.1-70B  | Tulu3 + longcite |      **15 min**          |        **5 sec**          |   1400 min      |     **1 %**        |
>
>
> Compared to the total training time (168 min),  the overall overhead of **3.25 min** is **negligible**. Note that HBP **ONLY** operates **before** the training process, so it introduces **NO** overhead during the training process.
>
> **Q3:**
> >In Section 4.3, the paper proposes the Stable Loss Normalizer and empirically selects the Average Token method, but lacks an in-depth analysis of the underlying reasons. Furthermore, it remains unclear whether the stability advantages of the Average Token normalizer apply exclusively to HBP, or if they also generalize to other methods. Could the authors clarify this?
>
> **A3**
>
> Thank you for your valuable feedback. We will clarify this in detail.
>
> * **Stable Loss Normalizer is not empirical**
>
> The Stable Loss Normalizer is designed based on theoretical principles. Its core purpose is to **ensure each token contributes equally to the overall loss**, removing biases from sequence length, data-parallel size, and gradient accumulation. Further analysis supporting this design is provided below.
>
>
> * **Can generalize to other methods**
>
> We conducted experiments on the loss normalizer under the base packing setting without HBP. The results show its effectiveness across different methods.
>
> |      Model        |      Dataset      | Loss Normalizer |  AVE  | LongBench |
> |:-----------------:|:----------------:|:---------------:|:-----:|:---------:|
> | LLama3.1-8B-ISF   | Tulu3 + longcite |   token-mean    | 56.0  |   44.0    |
> | LLama3.1-8B-ISF   | Tulu3 + longcite |  sample-mean    | 55.4  |   43.0    |
> | LLama3.1-8B-ISF   | Tulu3 + longcite |   **Ave-Token (Stable Loss)**     | **57.0**  |   **43.8**    |
>
> ---
>
> **In-depth analysis**
>
> To clarify, let’s consider a data-parallel (dp) group with a size of 2, where each rank runs with a local batch size of 2.
>
> Let $loss$ denote the sum loss for a sample, and $T$ represent the number of tokens involved in the loss calculation.
>
>
>   -  **Token-Mean**：
>
> In the token-mean approach, we compute the loss for each DP rank independently as follows:
>
>    $$
>    L_{dp_{1}} = \frac{loss_{1} + loss_{2}}{T_{1} + T_{2}}, \quad L_{dp_{2}} = \frac{loss_{3} + loss_{4}}{T_{3} + T_{4}}
> $$
>
> The final loss for the global batch is calculated by averaging the individual DP rank losses:
> $$L_{final}=\frac{L_{dp_{1}}+L_{dp_{2}}}{2} =\frac{loss_{1}+loss_{2}}{2·(T_{1}+T_{2})} + \frac{loss_{3}+loss_{4}}{2·(T_{3}+T_{4})}$$
>
> This method may introduce bias if the data-parallel ranks have different sequence length distributions, as it normalizes the losses independently for each rank **without considering the global batch’s overall token length distribution**.
>
> ---
>  - **Sample-Mean:**
>
> In the sample-mean method, losses for each DP are first normalized by their respective token lengths and then averaged across the samples:
> $$L_{dp_{1}}=\frac{\frac{loss_{1}}{T_{1}}+\frac{loss_{2}}{T_{2}}}{2} , \quad                   L_{dp_{2}}=\frac{\frac{loss_{3}}{T_{3}}+\frac{loss_{4}}{T_{4}}}{2}$$
>
> The final loss is then:
> $$L_{final}=\frac{L_{dp_{1}}+L_{dp_{2}}}{2} =\frac{loss_{1}}{4·T_{1}} + \frac{loss_{2}}{4·T_{2}}+\frac{loss_{3}}{4·T_{3}} + \frac{loss_{4}}{4·T_{4}} $$
>
> This method balances the contribution of each individual sample. However, it does not fully account for the total number of tokens in the global batch, resulting in **samples with fewer tokens having a disproportionately larger influence on the overall loss**.
>
> ---
>
> -  **Stable Loss Normalizer:**
>
> The Average Token normalizer ensures that the total number of tokens in a global batch is considered. Here, we define the average token length across all samples:
>
> $$T_{ave}   = \frac{T_{1} + T_{2} + T_{3} + T_{4}}{4}$$
>
> Each DP's loss is then normalized by  $T_{ave}$
>
> $$L_{dp_{1}}=\frac{loss_{1}+loss_{2}}{2·T_{ave}}, \quad       L_{dp_{2}}=\frac{loss_{3}+loss_{4}}{2·T_{ave}}$$
>
> The final loss is:
> $$L_{final}=\frac{L_{dp_{1}}+L_{dp_{2}}}{2} =\frac{loss_{1} + loss_{2} +loss_{3} + loss_{4}}{4·T_{ave}} = \frac{loss_{1} + loss_{2} + loss_{3} + loss_{4}}{T_{1} + T_{2} + T_{3} + T_{4}}$$
>
> This approach effectively normalizes the loss by the total number of tokens across all samples in a global batch, **ensuring that each token, regardless of which sample it belongs to, contributes equally to the final loss**.
>
> We hope our response addresses your concerns, and we welcome any further questions or feedback.

---

> > ### Author Response · Authors · 2025-08-07
> >
> > Dear Reviewer,
> >
> > Thank you very much for your thoughtful feedback and for taking the time to review our work.
> >
> > If any points in our response require further clarification, we would be happy to provide more details.
> >
> > Could you please let us know if our rebuttal has addressed your concerns, or if you have any additional questions or suggestions?
> >
> > Thank you again for your valuable time and input.

---

> ### Comment · Area_Chair_m4WA · 2025-08-07
>
> Hi Reviewer Nqmr, just following up to see if you have any additional feedback on the authors’ response. Feel free to check whether your concerns were addressed or if there are any further issues.

---

### Decision · Program_Chairs · 2025-09-17

**Decision:**

Accept (poster)

**Comment:**

This paper proposes a data packing method to address training efficiency issues for long-context large language models (LLMs).The authors propose Hierarchical Balance Packing (HBP), which introduces hierarchical group auto-selection,  balance packing, and  a dynamic training pipeline with curriculum learning and a stable loss normalizer. Experiments on models ranging from 8B to 236B parameters demonstrate consistent 1.3–2.4× training speedups with negligible or positive impact on downstream performance. The rebuttal effectively addressed efficiency and generalization concerns, strengthening the case for acceptance. Remaining limitations include heuristic design choices and clarity of presentation. Overall, given the importance of the problem, the empirical gains, and the authors’ clarifications, the paper leans toward borderline accept.